# Neuroelectrophysiology-compatible electrolytic lesioning

**Iliana E Bray[1†], Stephen E Clarke[2†], Kerriann M Casey[3], Paul Nuyujukian[1,2,4,5,6]\*, for the Brain Interfacing Laboratory**

[1]Department of Electrical Engineering, Stanford University, Stanford, United States; [2]Department of Bioengineering, Stanford University, Stanford, United States; [3]Department of Comparative Medicine, Stanford University, Stanford, United States; [4]Department of Neurosurgery, Stanford University, Stanford, United States; [5]Wu Tsai Neuroscience Institute, Stanford University, Stanford, United States; [6]Bio-X, Stanford University, Stanford, United States

**Abstract** Lesion studies have historically been instrumental for establishing causal connections between brain and behavior. They stand to provide additional insight if integrated with multielectrode techniques common in systems neuroscience. Here, we present and test a platform for creating electrolytic lesions through chronically implanted, intracortical multielectrode probes without compromising the ability to acquire neuroelectrophysiology. A custom-built current source provides stable current and allows for controlled, repeatable lesions in awake-behaving animals. Performance of this novel lesioning technique was validated using histology from ex vivo and in vivo testing, current and voltage traces from the device, and measurements of spiking activity before and after lesioning. This electrolytic lesioning method avoids disruptive procedures, provides millimeter precision over the extent and submillimeter precision over the location of the injury, and permits electrophysiological recording of single-unit activity from the remaining neuronal population after lesioning. This technique can be used in many areas of cortex, in several species, and theoretically with any multielectrode probe. The low-cost, external lesioning device can also easily be adopted into an existing electrophysiology recording setup. This technique is expected to enable future causal investigations of the recorded neuronal population's role in neuronal circuit function, while simultaneously providing new insight into local reorganization after neuron loss.

## eLife assessment

This paper reports a **valuable** new method for creating localized damage to candidate brain regions for functional and behavioral studies. The authors present **solid** support for their ability to create long-term local lesions with mm spatial resolution. The paper is likely to be of broad interest to brain researchers working to establish causal links between neural circuits and behavior.

## Introduction

Neuroelectrophysiology – using electrodes to record the electrical signals generated by neurons – has been the defining technique that shifted neuroscience from macroscale anatomy and loss-of-function studies down to the microscale activity of individual neurons and synapses in a given region of interest (*Sporns, 2016*). Prior to the last few decades, electrophysiology was limited to simultaneous recordings of at most five neurons with small numbers of electrodes (typically one, two, or four) (*McNaughton et al., 1983*). The development of neuroelectrophysiology recording techniques with a large number of electrodes starting in the 1970s (*Wise et al., 1970*; *Campbell et al., 1991*; *Campbell*

**\*For correspondence:**
24elife@pn.stanford.edu

[†]These authors contributed equally to this work

**eLife digest** Over the past three decades, the field of neuroscience has made significant leaps in understanding how the brain works. This is largely thanks to microelectrode arrays, devices which are surgically implanted into the outermost layer of the brain known as the cortex. Once inserted, these devices can precisely monitor the electrical activity of a few hundred neurons while also stimulating neurons to reversibly modulate their activity.

However, current microelectrode arrays are missing a key function: they cannot irreversibly inactivate neurons over long-time scales. This ability would allow researchers to understand how networks of neurons adapt and re-organize after injury or during neurodegenerative diseases where brain cells are progressively lost.

To address this limitation, Bray, Clarke, et al. developed a device capable of creating consistent amounts of neuron loss, while retaining the crucial ability to record electrical activity following a lesion. Calibration tests in sheep and pigs provided the necessary parameters for this custom circuit, which was then verified as safe in non-human primates. These experiments demonstrated that the device could effectively cause neuron loss without compromising the recording capabilities of the microelectrode array.

By seamlessly integrating neuron inactivation with monitoring of neuronal activity, scientists can now investigate the direct effects of such damage and subsequent neural reorganization. This device could help neuroscientists to explore neural repair and rehabilitation after brain cell loss, which may lead to better treatments for neurodegenerative diseases. In addition, this technique could offer insights into the interactions between neural circuits that drive behavior, enhancing our understanding of the complex mechanisms underlying how the brain works.

*et al., 1990*; *Rios et al., 2016*; *Hong and Lieber, 2019*) was a boon to systems neuroscience, enabling simultaneous recording from hundreds of neurons and revealing previously unseen aspects of population coding (*Cunningham and Yu, 2014*; *Kalaska, 2019*; *Shenoy and Kao, 2021*). Over the last five years, electrode count has continued to increase, with new probes containing thousands of electrodes (*Jun et al., 2017*; *Steinmetz et al., 2021*; *Obaid et al., 2020*; *Sahasrabuddhe et al., 2021*; *Musk, 2019*). While experiments and analysis have revealed population activity that correlates strongly with behavioral output (*Yu et al., 2009*; *Churchland et al., 2012*; *Sadtler et al., 2014*; *Golub et al., 2018*; *Elsayed et al., 2016*; *Goldman et al., 2019*; *Gallego et al., 2020*; *Liu et al., 2020*; *Rasmussen et al., 2017*), novel tools that both record from and inactivate neurons are required to establish causal connections to behavior (*Vaidya et al., 2019*; *Shenoy and Kao, 2021*; *Slonina et al., 2022*). Although many inactivation methods exist, to date it has been challenging to find a repeatable, long-term inactivation technique compatible with chronic intracortical neuroelectrophysiology.

## Design considerations and existing inactivation methods

Three main design considerations are required for a technique that successfully combines electrophysiology with inactivation (further detailed in Appendix 1).

1. Stable electrophysiology pre- and post-inactivation: Avoiding physical disruption enables direct comparison of pre-lesion activity with both the acute and chronic stages of injury.
2. Ability to localize and control the size of the inactivation: Precise focal inactivations strike a balance between being large enough to alter performance but small enough to spare sufficient tissue to record local re-organization and recovery.
3. Cross-compatibility: A technique that can be used in many areas of cortex, with any multielectrode probe, and in several species will enable causal investigation across a large variety of contexts. A method that works across-species (especially in large-animals like rhesus macaques) would leverage the existing injury literature, recording technologies, and behavioral assays from rodents to new world monkeys, while also being well-suited for the behavioral sophistication and human homology of macaques (*Higo, 2021*).

Neuronal activity can either be temporarily or permanently inactivated, defined as manipulation or termination, respectively (*Vaidya et al., 2019*). For clarity, termination refers to a technique that causes death of neurons, removing them from the circuit. We non-exhaustively review existing manipulation

and termination techniques in Appendix 2. However, none of these existing techniques meet all the above design considerations.

Manipulations enable a causal understanding of the relationship between neuronal activity and behavior by studying adaptation to and from a perturbation (*Slonina et al., 2022*). Existing manipulation methods include intracortical microstimulation, optogenetics, pharmacology, transcranial stimulation, cooling loops, and chemogenetics. Since neurons remain anatomically and physiologically viable, manipulations can be easily repeated, until desensitization occurs.

As termination causes cell death, it can generate stronger causal evidence than a manipulation (*Vaidya et al., 2019*). For example, even though a transient manipulation of neuronal activity may temporarily disrupt behavior, only a sustained manipulation could elicit the system's long-term adaption (*Slonina et al., 2022*). Over the days (*Ferrier and Yeo, 1884*) and potentially weeks *Bundy and Nudo, 2019*; *Zeiler et al., 2016* following a termination, the surrounding circuitry and broader network may adapt, leading to behavioral recovery and demonstrating that although the terminated region was causally implicated in behavioral control in the moment, the terminated neurons are *not themselves* causally necessary. A sustained manipulation could accomplish a similar effect to a termination at the systems level, up to the point at which the manipulation stopped. However, it can be difficult to create sustained inactivation with existing reversible manipulation methods, limiting their ability to study the brain's natural reorganization over timescales of days to months.

Termination methods overcome the problem of sustained and consistent inactivation from which temporary inactivation techniques suffer. Therefore, they enable a form of causal inference not possible with temporary inactivation methods (*Vaidya et al., 2019*; *Shenoy and Kao, 2021*). Existing termination techniques include mechanical trauma, endovascular occlusion, Rose Bengal mediated photothrombosis, and chemical lesioning. Prior termination studies mostly measure behavioral output, with no simultaneous measures of neuronal activity during the behavior, impairing their ability to provide insight into the causal connection between the brain and behavior (*Morissette and Boye, 2008*; *Nudo et al., 2003*; *Nudo, 2013*; *Wurtz and Goldberg, 1972*; *Glees and Cole, 1950*), or with no baseline (i.e. pre-lesion) measures of neuronal activity (*Khanna et al., 2021*).

## Electrolytic lesioning through a microelectrode array

In order to best meet the three design considerations, we created a device to make an electrolytic lesion through the same microelectrode array used to record neuronal activity. The dual use of this microelectrode array achieves the first design consideration of stable electrophysiology pre- and post-inactivation, as it enables recording from the exact same signal source with minimal physical disruption or displacement. Aside from the initial implantation of the array, there are no invasive procedures required, removing virtually all risk of destabilizing the recorded population. A further benefit of this surgery-free procedure is avoiding analgesia and sedation and allowing for an unprecedented, minutes-long turn-around between pre- and post-lesion data collection; the experiment can resume as soon as close observation of the animal is complete, the lesion device is disconnected, and the recording stream is reconnected. The electrolytic lesioning technique is repeatable, because the same multielectrode probe can be used many times to create lesions while maintaining stable electrophysiological recordings. Performing electrolytic lesioning requires passing a specified amount of current through two electrodes (one acting as the anode, the other as the cathode). Selecting which electrodes in the microelectrode array should be the anode and cathode sets the location of the lesion origin. By using a Utah multielectrode array as in this work, the location of the lesion can be changed at a 400 μm resolution set by the electrode spacing, while altering the duration and amount of current passed through two electrodes creates changes in the lesion's spatial extent. Thus, while there is still variation in the precise geometry of damage from each lesion, multielectrode-based lesioning does well to satisfy the second design consideration and is a notable refinement from previous lesioning studies. While this platform was designed for use in rhesus macaques, it could be used in other animals in which multielectrode probes can be implanted, such as other primates, large mammals, and rodents. This technique can also be used in effectively any area of cortex in which a multielectrode probe could be implanted. While the link between the lesion location and the multielectrode location technically constrains the lesion to an area of cortex in which a multielectrode array could be implanted, we see the connection as a positive, because it ensures recording some neuroelectrophysiology from the perilesional area in which recovery is hypothesized to occur (see Appendix 1).

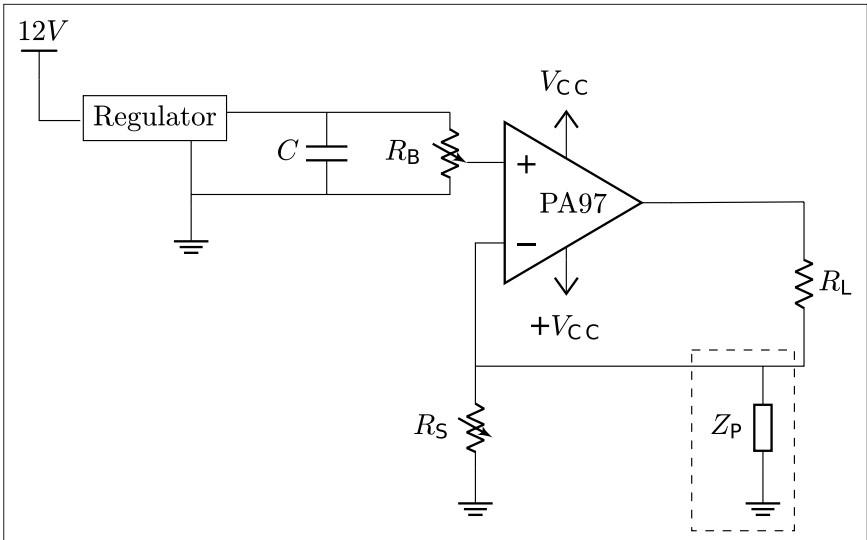

**Figure 1.** The circuit diagram for the electrolytic lesioning device. An op-amp is used in a negative feedback loop to maintain a constant current through the two electrodes in the brain ($R_L$). The op-amp was implemented as suggested by its accompanying evaluation kit and supplied components. The system is powered by a 12 V power supply, and a boost converter is used to create a $V_{CC}$ and -$V_{CC}$ of 450 V and –450 V, respectively. The current through $R_L$ can be set by changing the resistance of the potentiometer, $R_S$. $Z_P$ is a hypothesized physiological parasitic component, which could be either resistive or capacitive (dashed box).

The online version of this article includes the following figure supplement(s) for figure 1:

**Figure supplement 1.** Connection diagram of the experimental setup for creating electrolytic lesions.

Therefore, this platform is compatible across many cortical areas, across several species, and theoretically across multielectrode probes, fulfilling the third design consideration. As the lesioning device is a small, low-cost external system that only needs to be connected for the duration of lesioning, it is easy to adopt into existing electrophysiology recording settings. This platform is anticipated to facilitate causal explorations of the relationship between brain and behavior through its unique combination of inactivation and neuroelectrophysiology. Further, it should enable studies of natural reorganization at the neuronal population level.

## Results

### Electrolytic lesioning device and testing

We created a custom current source circuit which allows us to use the same microelectrode array used to record neuronal activity to also create an electrolytic lesion (*Figure 1*). The circuit design is based on simple feedback control circuits *Texas Instruments Incorporated, 2019*; *Carter and Brown, 2016*; *Texas Instruments Incorporated, 2020* using an operational amplifier to maintain a constant current (*Carter and Mancini, 2009*). While this technique should theoretically work with any multielectrode probe, we performed all proof-of-concept experiments with a Utah microelectrode array (Blackrock Neurotech, Salt Lake City, UT). Lesion size is controlled by the amplitude and duration of the current passed through the two chosen electrodes of the microelectrode array, although the exact extent of damage will vary for each lesion due to anatomical variation. Our circuit supplies the voltage needed to damage cortex while maintaining high precision in the delivered current (precision error not exceeding 10 µA). To find a combination of the current amplitude and duration parameters that would create a reasonable lesion size for our use in rhesus macaques, testing was first performed in ex vivo sheep and pig brains, and then in vivo with anesthetized pigs.

Although scientific use of this lesioning device is ultimately intended in rhesus, pilot studies were performed in sheep and pigs in accordance with the guidelines of the *National Research Council, 2003*; *National Research Council, 2011* of replacement, refinement, and reduction (3 R's). Cortex was lesioned using the chosen parameters, after which histology was collected to provide an

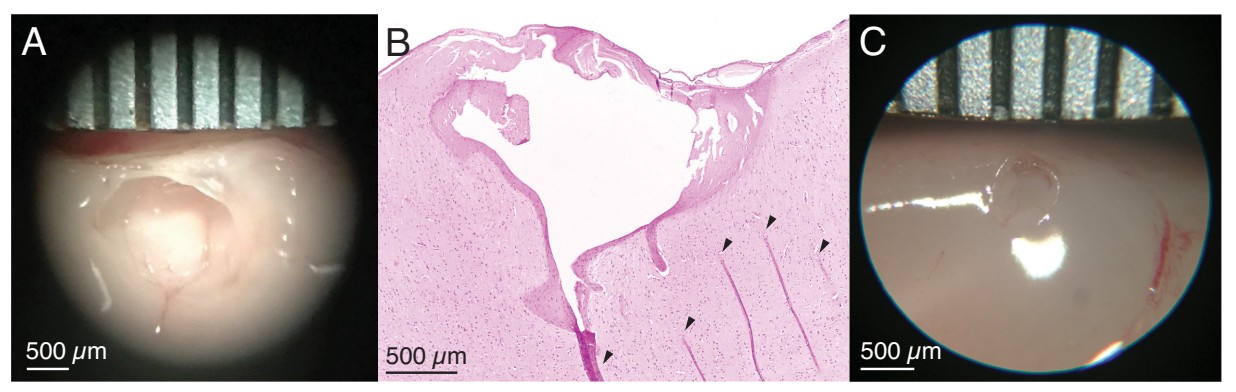

**Figure 2.** Ex vivo testing to calibrate lesion parameters. (**A**) Ex vivo demonstration of the electrolytic lesion technique in unfixed sheep cerebral cortex using an intracortical Utah microelectrode array. Sustained delivery of 250 µA of direct current for 10 min between adjacent electrodes (400 µm spacing) resulted in a clean spheroidal cavitation in cortex approximately 1.5 mm in diameter. Ruler is marked every 500 µm. (**B**) Hematoxylin and eosin (H&E) stained slice of the lesion in (**A**) clearly shows the lesioned region. Arrows indicate tissue fold artifacts that resulted from the histology process, not the lesion. The other dark pink areas surrounding the cavitation in cortex are regions of necrosis. (**C**) A smaller ex vivo lesion in unfixed cerebral cortex of a pig created by decreasing the direct current amplitude and duration to 180 µA for 1 min. The cavitation has a diameter slightly over 0.5 mm.

The online version of this article includes the following figure supplement(s) for figure 2:

**Figure supplement 1.** Locations of the electrodes used to lesion and the relative size of the lesion area to the array area for the testing in **Figure 2**.

understanding of lesion quality and extent. Our exploration was not exhaustive and not designed to fully characterize lesion extent as a function of current amplitude and duration; while being mindful of animal use, a key aim was to converge on a parameter set that yielded small, focal lesions suitable for behavioral studies in rhesus macaques with chronic implants. We both sampled the search space of parameter values, current amplitude and duration, and repeated parameter selections for confirmation. Across eleven ex vivo and five in vivo animals, 61 lesions were performed, including some repeated parameter combinations as biological replicates.

## Ex vivo ovine and porcine testing

Initial testing was performed in unfixed cortex from sheep and pigs. Prior studies demonstrated that 20 min of 400 µA direct current resulted in a spherical cavitation in cortex of approximately 2 mm in diameter (**Wurtz and Goldberg, 1972**). To create a more localized lesion, parameters were reduced substantially to 250 µA direct current for 10 min, passed through two adjacent electrodes of a Utah array implanted in ex vivo sheep cortex. This created a clean spheroidal cavitation in cortex approximately 1.5 mm in diameter (**Figure 2A and B** and **Figure 2—figure supplement 1A, B**), which is consistent with lesion sizes that cause measurable behavioral deficits in the motor system (**Nudo et al., 2003**; **Wurtz and Goldberg, 1972**; **Glees and Cole, 1950**). Although lesions of this size are known to cause significant behavioral deficits, we hypothesized that smaller lesions might lead to deficits that were still noticeable but not detrimental to the animal's overall mobility and mental health, a noted concern in past studies (**Glees and Cole, 1950**). Reducing the duration and intensity of the current used to lesion created smaller cavitations in cortex: 1 min of 180 µA direct current, passed through two adjacent electrodes in ex vivo pig cortex, created a smaller spheroidal cavitation of approximately 0.5 mm in diameter (**Figure 2C** and **Figure 2—figure supplement 1C, D**).

## In vivo porcine testing

In vivo testing allowed for further parameter refinement in the presence of natural acute inflammatory responses. By further reducing the duration and intensity of the current used to lesion, the extent of the cavitation in the cortex was reduced to the point where there is no longer a cavitation but instead a clear region of damaged parenchyma. One minute of 150 µA direct current, passed through two adjacent electrodes in pig cortex resulted in a well-demarcated region of parenchymal damage (**Figure 3A** and **Figure 3—figure supplement 1**). The region of parenchymal damage appears paler on H&E staining than the adjacent, unaffected parenchyma and was confined to an upside-down

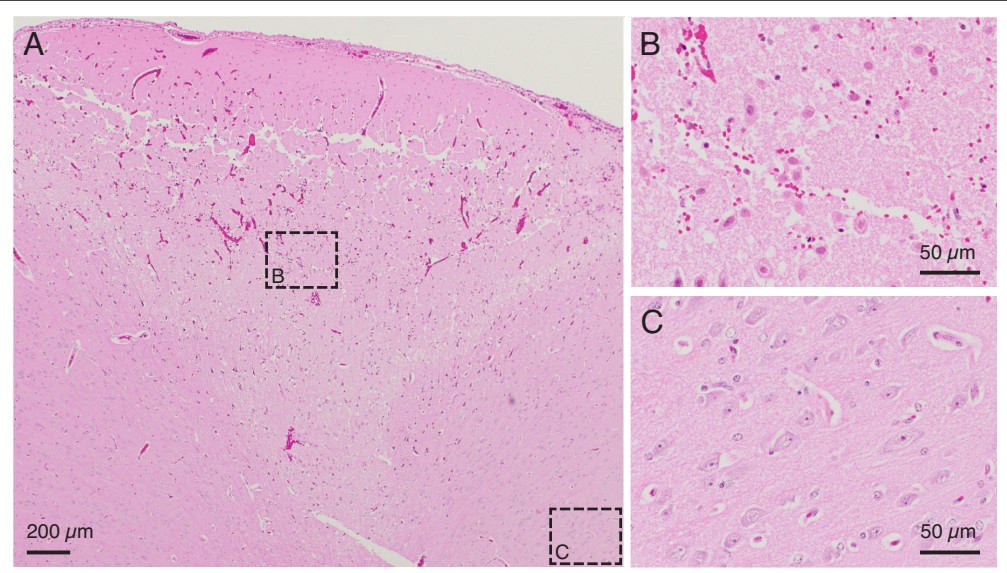

**Figure 3.** In vivo testing to further calibrate lesion parameters. (**A**) H&E stained slice from an in vivo demonstration of the lesioning technique in pig cerebral cortex. 150 µA direct current passed through two adjacent electrodes (400 µm spacing) for 1 min resulted in a conical region of damaged parenchyma. The top of the conical region shows a line of damage which may be caused by physical removal of the microelectrode array after testing. Anatomically observed alterations are clearly demarcated, emphasizing the fine localization of the lesioning method. Note: An outline of the region of damaged parenchyma is shown in *Figure 3—figure supplement 1*. (**B**) Region of intermixed necrotic and histologically normal neurons within the conical zone of damage is visible in a close-up of the slice from (**A**). Necrotic neurons have shrunken cell bodies. The microelectrode array is expected to continue recording from remaining healthy neurons after performing a lesion. (**C**) Region of viable neurons outside the conical region of damage is visible in a close-up of the slice from (**A**). This shows the precise spread of the method, with intact, viable tissue present just outside the lesioned area.

The online version of this article includes the following figure supplement(s) for figure 3:

**Figure supplement 1.** The H&E stained slice from *Figure 3A*, with the conical region of damaged parenchyma outlined with a dashed white line for clarity.

conical region, which was 3.5 mm wide at the cortical surface and extended approximately 2 mm deep. Within this region (seen magnified in *Figure 3B*), there was widespread coagulative necrosis, parenchymal rarefaction, and perivascular microhemorrhage. Acidophilic neuronal necrosis was identified along the marginal boundaries of the region of coagulative necrosis. Adjacent neural parenchyma was unaffected (representative selection shown in *Figure 3C*), demonstrating the relatively precise boundary of damage caused by the electrolytic lesion. The physical damage visible as tears in the tissue (white) near the surface of this damaged cone of tissue may be due to withdrawal of the microelectrode array from cortex after testing. In control tests where a microelectrode array was inserted and removed but no current was used to create a lesion (*Supplementary file 1* and *Supplementary file 2*), regional coagulative necrosis was not present, and cortical damage was confined to mild subcortical and/or perivascular microhemmorhage and scattered individual neuronal necrosis, typically in regions adjacent to microvascular hemorrhage. This emphasizes that electrolytic lesioning, not array insertion alone, leads to coagulative necrosis. This testing was performed across many regions of porcine cortex, demonstrating that the electrolytic lesioning technique functions in any area of cortex in which a multielectrode probe can be implanted.

## In vivo use in rhesus macaques

After this validation and refinement, one proof-of-concept lesion (150 µA direct current passed through adjacent electrodes for 45 s) was performed in an in vivo sedated rhesus macaque (Monkey F) in order to validate the safety of the procedure. This technique has since been used successfully in

experimental settings with two other awake-behaving rhesus macaques (Monkeys H & U), for a total of fourteen lesions using thirteen unique electrode pairs.

## Relationship between applied current and lesion volume

Our ex vivo ovine and porcine testing and our in vivo porcine testing demonstrated a relationship between the amplitude and duration of the direct current applied and the size of the lesion. In order to characterize this relationship, we compared lesion volume estimates for a variety of direct current amplitudes and durations. We had slightly fewer data points for the in vivo lesions, due to difficulty in obtaining histology slices that captured the damage from the lesion within the slice. Lesion volumes were estimated from histology or photographs taken from our ex vivo and in vivo testing. Volumes were estimated using volume formulas for the approximate geometries observed visually in the histology. Most volumes were estimated assuming a conical lesion volume, chosen based on *Figure 2* and *Figure 3*, though one in vivo volume was visually identified as spherical and estimated as such. We found that as both the amplitude and duration of direct current increase, the volume of the lesion also increases (*Figure 4*). For ex vivo lesions with a current amplitude of 180 µA, we found that volume was an exponential function of the duration of current (*Figure 4B*, $V = \left( 5.4 \times 10^{-7} \right) \times e^{5.6t}, R^2 = 0.413$). We also found that as amplitude and duration of direct current decrease, the type of damage changes (*Figure 5*). While lesions with larger current amplitudes and durations create cavitations in cortex (as in all of the ex vivo and the longest in vivo lesion), lesions where the current had a lower amplitude and/or was applied for a shorter duration resulted in rarefied tissue damage (as seen in *Figure 3*), containing parenchymal rarefaction, coagulative necrosis, and perivascular microhemorrhage.

## Current and voltage output

In all of the fourteen lesions across two awake-behaving rhesus macaques (150 µA direct current passed through adjacent electrodes for 30 or 45 s [30 s for Monkey U and 45 s for Monkey H, except lesion H200120 which was for 50 s]), the current source worked as expected, providing a constant current throughout the duration of the procedure. Fluctuations in current amplitude were likely due to the 10 µA precision of the multimeter used to read out the current (e.g. the readout would sometimes switch between 150 µA and 160 µA). The voltage across the microelectrode array fluctuated much more than the current did, emphasizing that we made the correct choice in using a current source to ensure delivery of consistent amounts of current into the brain (*Figure 6*). Upon turning on the lesioning device, the voltage initially increased sharply, sometimes exceeding the slew rate of the voltmeter (seen as a discontinuity in the traces in *Figure 6*). Due to the limited resolution of the voltmeter, the voltages were unknown between 0.13 and 0.33 s but could not have exceeded 900 V based on $V_{CC}$ and -$V_{CC}$. After this peak, the voltage predominantly levels off.

Duration of the applied current is controlled by a switch that cuts the power to the boost converter. However, after the power to the boost converter is removed, the current supplied by the circuit briefly persists — likely due to residual energy present in the capacitors of the boost converter downstream of the switch. In the lesions for which this data was collected, the current persisted at the calibrated intensity for between 2.5 and 4 s.

Before performing a lesion, the amount of current to be output by was calibrated using a 50 kΩ resistor in place of the implanted microelectrode array and altering the resistance level of the external potentiometer. Even when calibrated to generate 150 µA, the actual current output when lesioning differed slightly (10–20 µA above or below the set value). In nine out of the fourteen lesions performed in two awake rhesus macaques (thirteen of which had this voltage data collected), the current value was higher than what was calibrated. We hypothesize that this is due to a parasitic parallel resistance to ground through the animal itself (see dashed box in *Figure 1*). To have created the 10–20 µA increase in current above the set value, this parasitic parallel resistance would have been between 1.2 MΩ and 0.6 MΩ, respectively, which is in the range of the expected resistance when the body is in dry contact with the environment (*Fish and Geddes, 2009*). In the three lesions where the current value was 10–20 µA lower than what was calibrated, there may have been some parasitic capacitance present. As these parasitic resistances and capacitances arise only when the animal is part of the experimental setup, and they change across sessions, they cannot readily be calculated and accounted for a priori. These parasitics appear small enough not to significantly impact the desired lesion characteristics.

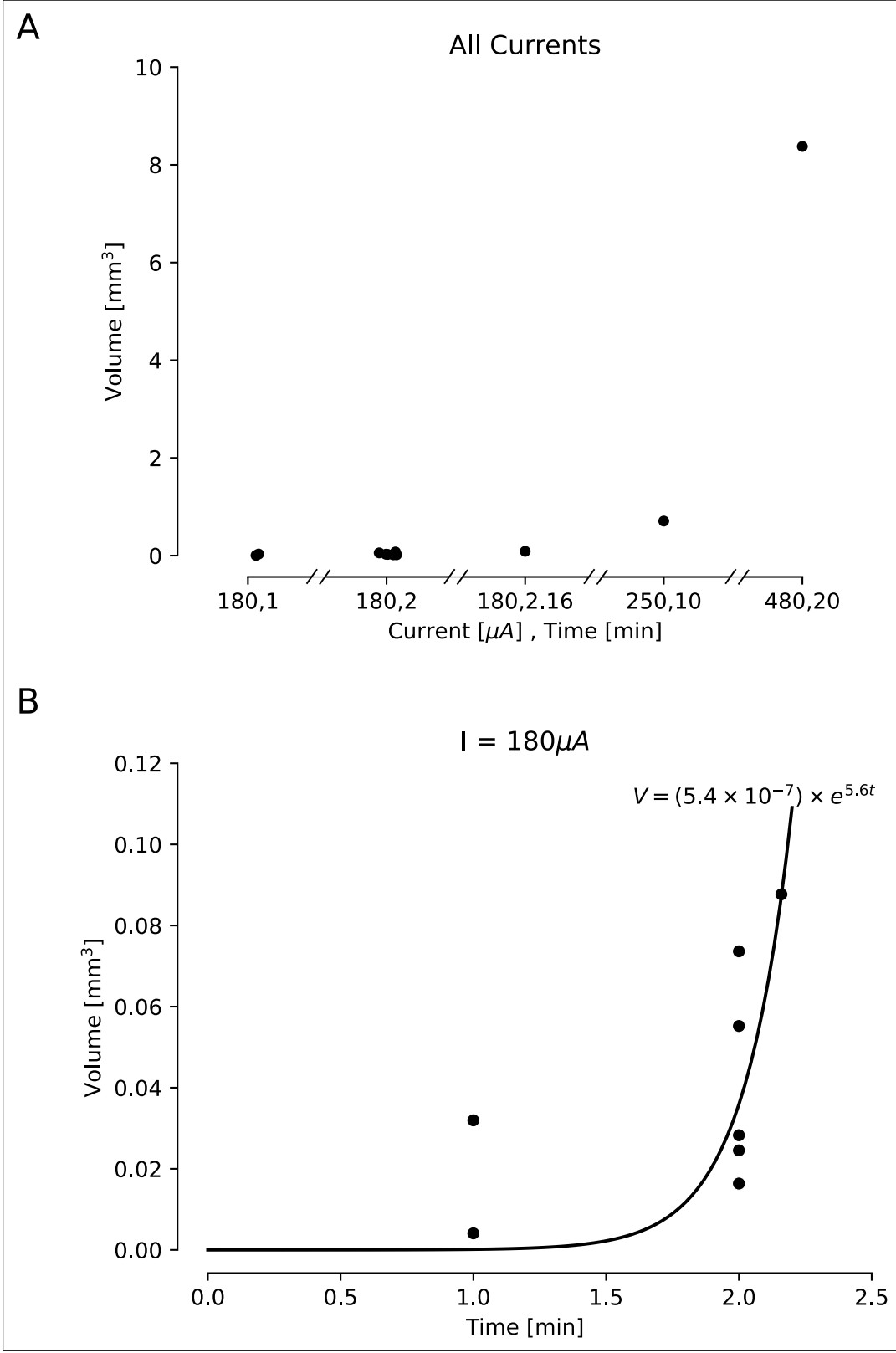

**Figure 4.** Estimated lesion volumes from lesions created in ex vivo sheep and pig cortex. All volumes were cavitations in cortex and were estimated assuming a conical lesion volume. (**A**) Estimated volumes are shown with black dots for each direct current amplitude and duration pairing. (**B**) Estimated volumes are shown for a variety of

*Figure 4 continued on next page*

*Figure 4 continued*

durations of applied 180 µA direct current. A curve was fit to this line, showing an exponential relationship between duration of the current and lesion volume ($V = \left(5.4 \times 10^{-7}\right) \times e^{5.6t}, R^2 = 0.413$).

## Recording quality

One advantage of microelectrode arrays is their ability to record from a stable population of cortical neurons over months (*Vaidya et al., 2014*; *Dickey et al., 2009*; *Fraser and Schwartz, 2012*; *Ganguly and Carmena, 2009*). While the exact neurons captured from the local population vary day-to-day, they largely remain the same, which enables researchers to sample the activity of a consistent neuronal population over time.

Comparisons of the recorded action potential waveforms before and after multiple lesions revealed that microelectrode arrays were able to continue recording stable neuronal activity in awake-behaving rhesus (e.g. after one lesion performed in Monkey U and after six lesions performed in Monkey H; *Figure 7A*). Surprisingly, even the lesion electrodes themselves continued to record reliably, suggesting that the modest electrolytic lesion intensity used is not prohibitively destructive to an electrode's recording ability. Given the majority of waveforms appear similar (*Figure 7A*), it seems unlikely that any acute damage response physically shifted the array away from the recorded neuron population.

After an electrolytic lesion, a fraction of the recorded waveforms appeared to change significantly, which may be a result of neuron damage or death, or may be an adaptive neuronal circuit response

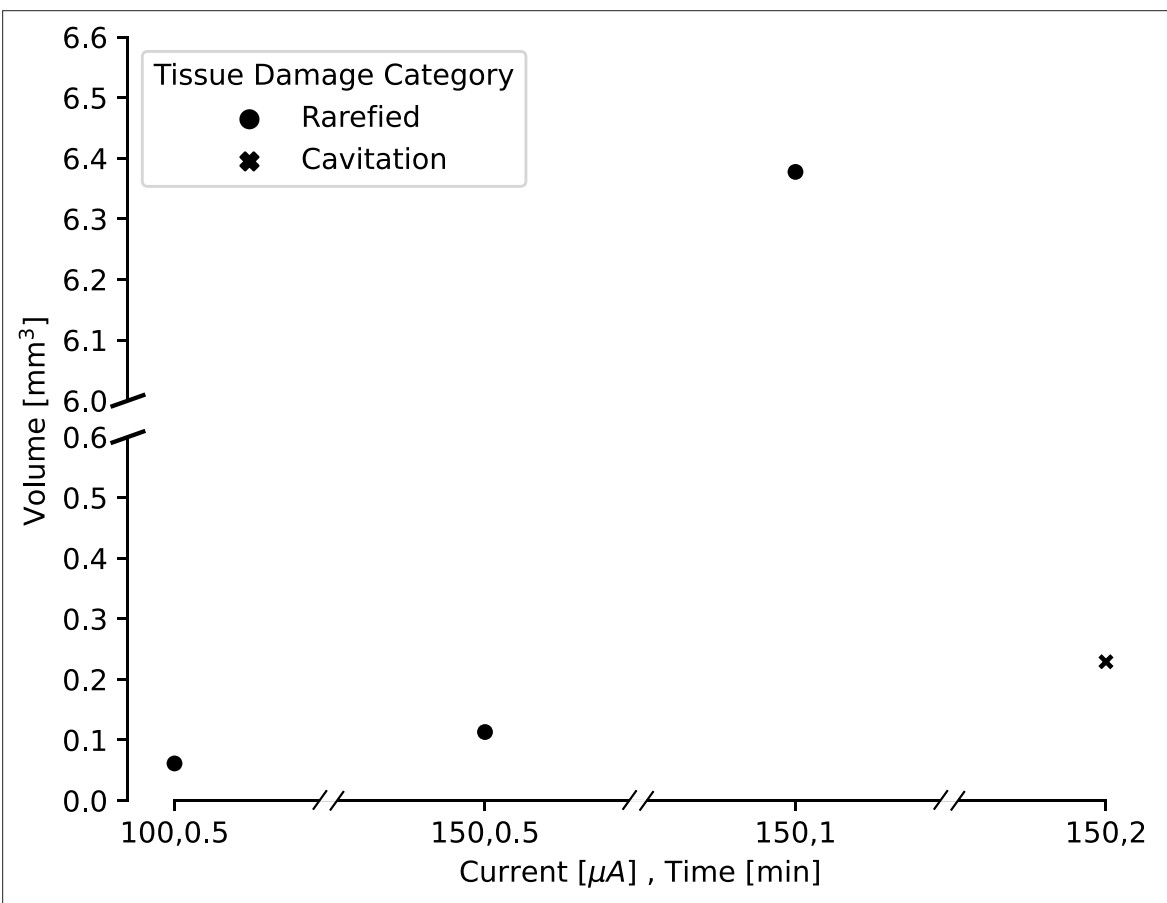

**Figure 5.** Estimated lesion volumes from lesions created in vivo pig cortex for a subset of direct current amplitude and duration pairings. For rarefied tissue damage, the lesion volume is indicated with a black dot, while for cavitation damage, the volume is indicated with a blue dot. Estimated lesion volumes were calculated from histology measurements and assumed a conical lesion volume (with the histology slice bisecting the volume), except for the lesion where 150 µA direct current was applied for 30 s, which was calculated as a spherical volume as indicated by the visually identified damage in the histology slice.

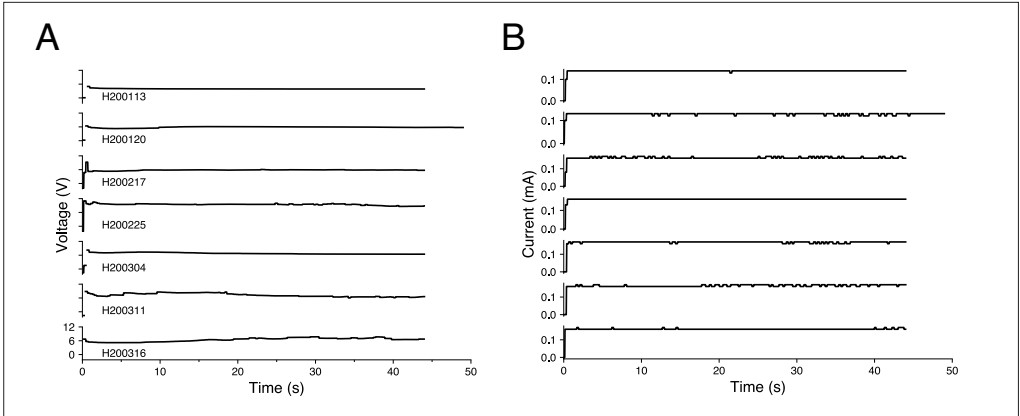

**Figure 6.** Voltage and current traces from seven representative lesions in an awake-behaving rhesus macaque (Monkey H). Lesions are shown in chronological order and are labeled with an experimental ID in the form SYYMMDD, where S indicates the animal, followed by the date. Traces only capture the values while the lesioning device was turned on (45 s for most lesions and 50 s for lesion H200120). (**A**) Voltage traces. Discontinuity at the beginning of the traces indicates transient voltages that were too rapid to be captured by the voltmeter, lasting between 0.13 and 0.33 s. The fluctuating voltages, especially the rapid increase in voltage at the beginning of lesioning, emphasize the importance of using a current source to deliver consistent amounts of current into the brain. (**B**) Current traces. The device delivered stable current for the duration of the lesion. Fluctuations are likely due to the 10 μA precision of the multimeter that read the current.

to lesioning. Since the lesion experiments performed in Monkeys H and U were not terminal studies, definitive histological evidence of neuron loss could not be acquired. Therefore, as a proxy for neuron loss, the relative change in the daily turnover of recorded neurons was assessed (**Figure 7B**; see Methods for details).

For each lesion, pairwise daily recording sessions were drawn from a set of thirteen contiguous recording days (four pre-lesion days and nine post-lesion days). For each electrode, principal component analysis was performed on standard normalized, sampled waveforms from the first of the 2 days; next, waveforms sampled from the second day were normalized and projected onto the top two dimensions determined from the first day (**Figure 7B**). A circle that captured half of the scatter points measures the spread, and changes in its radius were assumed to reflect changes in the recorded multi-unit activity. Typically, the difference in radius was nearly zero, reflecting similar proportions of activity from a similar composition of nearby neurons. Modest decreases and increases in radius were also frequently observed and the projected waveforms appeared similar. Two final categories were observed less commonly, in which the radius decreased or increased dramatically and the projected waveforms were substantially different.

Comparison days were grouped as pre-pre, pre-post, which included an acute period of up to 3 days following a lesion, and post-post, which compared post-lesion days 4 to 7 and was intended to reflect a late post period (not necessarily a complete recovery of brain and behavior). All pairwise comparisons were separated by no more than 4 days to minimize natural turnover rates described in the literature (**Gallego et al., 2020**). Following lesions in Monkey U (**Figure 7C**) and Monkey H (case 1; **Figure 7D**), a distinct cluster was detected in the histogram of radial changes ($\Delta r$) for all array electrode comparisons between pre-lesion and acute post-lesion days. This group captures the either complete silencing of neurons or their putative loss and appeared (significantly) in the pre-post group only. As a measure of turnover across the array, a % match was determined for each groups' pairwise comparison between daily recording sessions, which reflects the number of electrodes where $\Delta r$ remained unchanged. These values dropped below the expected percentage of matching neurons among pre- and post-lesion days (pre-pre, post-post) confirming that rates of turnover in the recorded population had been accelerated by the lesion. Although similar results were obtained for case 2 in Monkey H, they were not significant (**Figure 7D**). This is largely due much larger variability in turnover and the fact that these lesions in Monkey H were performed before complete recovery was observed.

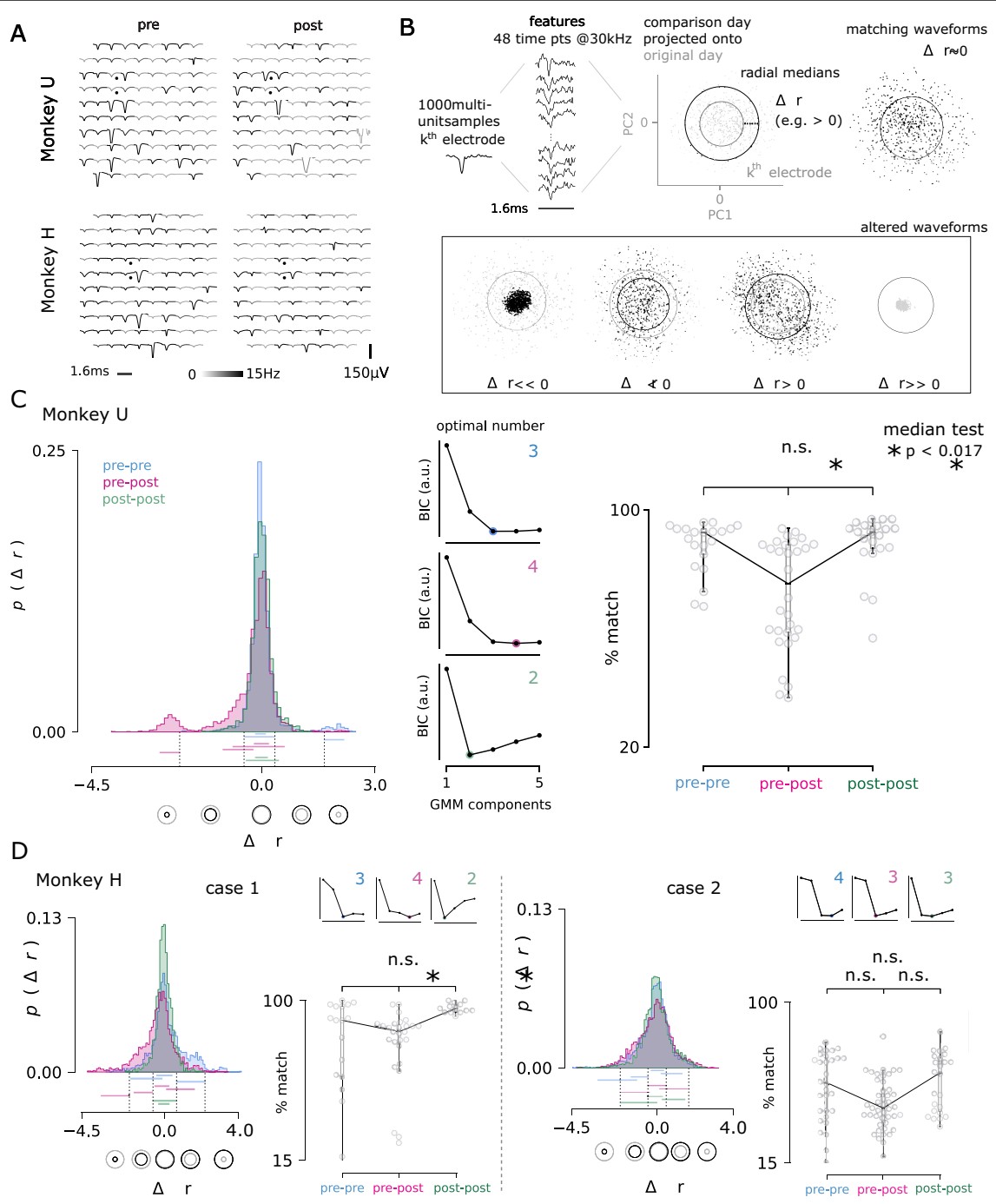

**Figure 7.** Electrolytic lesions perturb neuron populations while maintaining stable recordings. (**A**) A representative comparison of recorded action potential waveforms, before and after the first lesion in Monkey U (*top*) and the sixth lesion in Monkey H (*bottom*). The location of the lesion electrodes are marked by black dots, showing visible changes in waveforms with low spatial specificity across the array. Signal was observed in the recording sessions immediately before and after lesioning (*left, right*). An action potential detection rate was determined from both periods of rest and task engagement (gray-scale shading, capped at 5 Hz for visualization). (**B**) The multi-neuron activity recorded on each electrode was analyzed for changes in the proportions of activity. Action potential waveforms from a selected day (black) were compared to 1000 waveforms from a previous day (grey) by projecting the selected day's waveforms into the top two principal components determined from the previous day. The median radius of all waveform points relative to the origin in these two dimensions are represented as circles for both the comparison and original day. The difference in radii, $\Delta r$, was computed for each channel and for all pairs of recording days, separated by no more than four days to ensure minimal rates of spontaneous turnover noted in the literature **Gallego et al., 2020**. Changes in radius could arise through changes in the relative proportion of activity among the recorded neurons, as well as putative neuron loss ($r \ll 0$) and gain ($r \gg 0$). (**C**) The day-to-day pairwise comparisons fall within three groups: pre-lesion days

*Figure 7 continued on next page*

*Figure 7 continued*

(pre-pre), pre-lesion versus post-lesion days (pre-post; up to 3 days post-lesion), and post-lesion days (post-post; four to seven days after a lesion). In all comparison cases, 24, 48, 72, or 96 hr separated the recording sessions. Distribution of the $\Delta r$ values for all channels and days are shown for each group from Monkey U. *Central panels* Gaussian mixture models were fit to the data and the optimal number of components for each was determined using the Bayesian information criterion (BIC, normalized arbitrary units). An entirely new cluster is identified for the pre-post group that is hypothesized to largely represent the loss of neurons from the local population, beyond the usual rate of turnover observed in pre-lesion conditions. *Right panel* The three groups were then compared by looking at the percentage of the 96 electrodes that matched across comparison days. The percentage of matching neurons dropped significantly after a lesion (median test; * < 0.017, corresponding to the conservative Bonferroni corrected significance threshold of 0.05 for the three comparisons). (**D**) The same analysis is performed for Monkey H, which yielded two cases: lesions consistent with Monkey U (lesions 2, 4, and 7; *left*) and those with high levels of turnover before and after injury (lesions 5, 6, 8, 9, and 10; *right*). Note, lesions in Monkey U were well-spaced out over three months and considered as independent samples. Lesions in Monkey H were performed in much quicker succession, which likely contributes to the discrepancy.

## Use of electrolytic lesioning device with other multielectrode probes

In order to verify that this method for creating electrolytic lesions through a multielectrode probe works with different probe types, we performed tests using a Plexon U-Probe–a linear electrode with 24 contacts spaced 100 µm apart–in ex vivo rabbit cortex. Using our original electrolytic lesioning device and the U-Probe, we created five distinct lesions through contiguous contacts (100 µm apart) by applying differing current amplitudes (240 µA and 150 µA) for differing durations (30, 60, and 120 s). In *Figure 8*, we show the location of the five lesions, as well as the current and voltage traces (as in *Figure 6*). This testing demonstrates the ability to deliver steady current through a different multielectrode probe from the Utah microelectrode array, which was the primary probe used in most of these experiments.

## Discussion

In this report, we demonstrated a novel method for electrolytic lesioning through a microelectrode array that is compatible with electrophysiological recording of neuronal activity pre- and post-lesion. To achieve this, a custom current source was built that would ensure stable current delivery throughout the lesion, for repeated lesions, as well as across different electrode types and animals. This degree of control represents a significant refinement of lesion studies in systems neuroscience research. This model was tested ex vivo with porcine and ovine brains and in vivo with porcine brains. Altering the amount and duration of the current changed the size of the lesion, as evidenced by the histology. Selection of two adjacent lesion electrodes allows for spatial localization under the 16 mm² array. Following these preliminary tests, one lesion was performed in a sedated rhesus macaque to verify the safety of the procedure, and fourteen lesions were performed between two awake rhesus macaques. Readouts from the lesioning device itself (both current and voltage) and electrophysiology from the microelectrode array verify that the device delivers the desired power and does not damage the array's recording ability. We believe that this electrolytic lesioning technique will improve understanding of the motor system by coupling lesioning (an established termination technique) with the detailed spatiotemporal measurements of intracortical electrophysiology (*Shenoy and Kao, 2021*; *Vaidya et al., 2019*).

Although it shares elements with other widely used techniques, performing electrolytic lesioning through a microelectrode array is not a common inactivation technique. For example, injecting current into the brain through microelectrodes is commonly done in intracortical microstimulation (*Weiss et al., 2019*). However, typical current values for microstimulation are around three times smaller than for lesioning, pulse durations are on the order of tens of microseconds, and the pulses are charged balanced so they by design do not lead to cell death or materially affect the electrodes. Similarly, passing current through microelectrodes to mark their location in cortex is a well-established method for tracking the location of electrodes at the end of an animal study. When histology is eventually performed, the lesion is used to find the anatomical location where electrophysiology was conducted, and it is compatible with single or multi-channel electrodes (*Chen et al., 2009*). However, the intent is different, as electrolytic lesions for marking are not generally used as an inactivation technique. Electrolytic lesioning for inactivation was historically done through a single barrel electrode as the anode and without any post-lesion electrophysiology

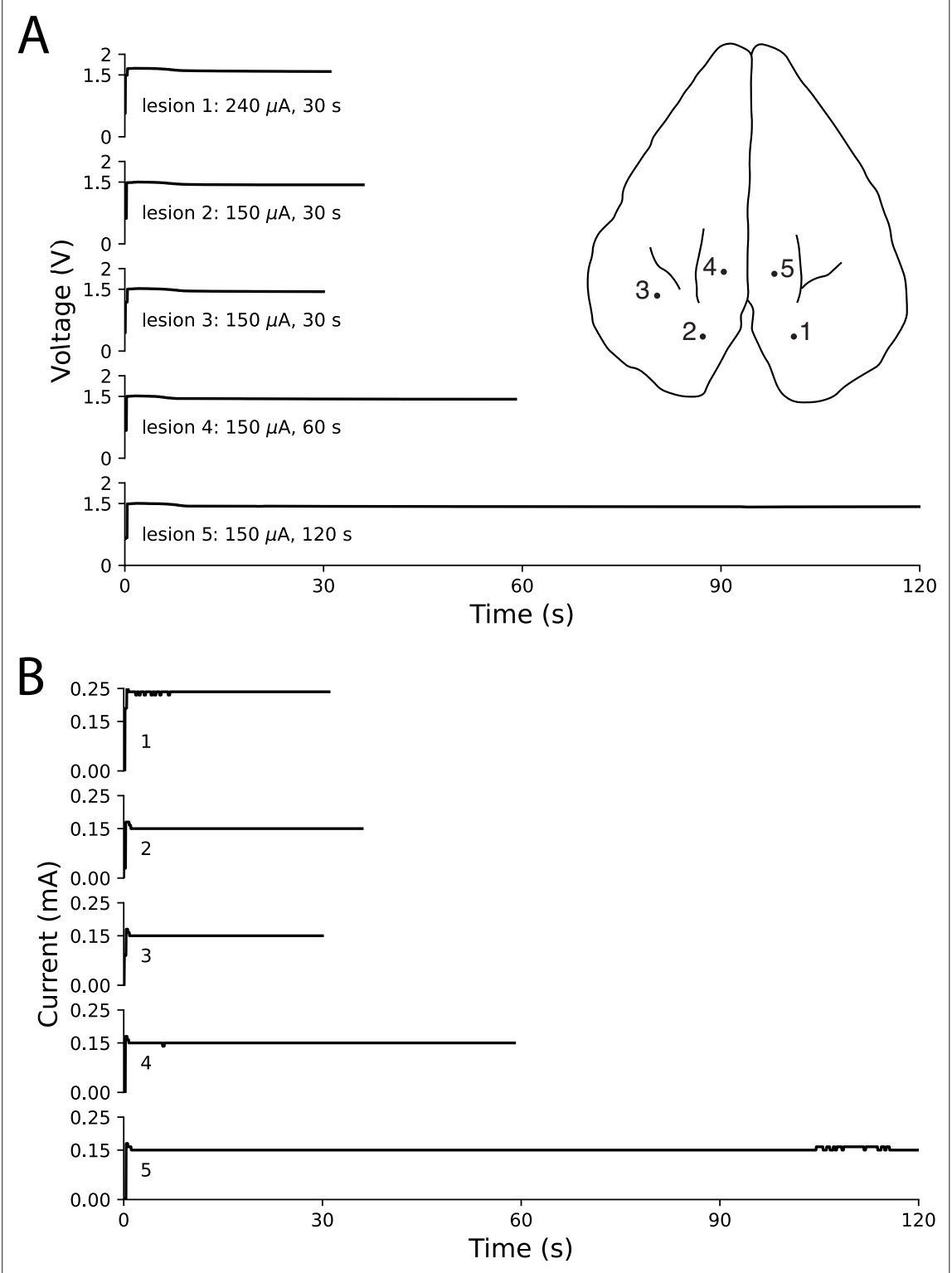

**Figure 8.** Ex vivo testing in rabbit cortex using a linear multielectrode probe. Surface penetration locations for the five lesions made with a linear multielectrode probe are marked on rabbit cortex, with corresponding voltage traces (**A**) and current traces (**B**) arranged top-to-bottom. Electrode contacts were separated by 100 μm and lowered approximately 400 μm below the surface. The voltage slowly decreased throughout each lesion, while the circuit maintained steady current delivery. Only minor fluctuations in current of less than 10 μA were observed, based on the resolution of the ammeter.

recordings (*Wurtz and Goldberg, 1972*). Therefore, the novel electrolytic lesioning method for inactivation presented here both is a refinement in precision and is unique in its compatibility with electrophysiology.

Electrolytic lesioning leads to cell death through heat (*Nikfarjam et al., 2005*), electroporation *Batista Napotnik et al., 2021*), and local changes in pH (*Nikfarjam et al., 2005*. Demonstration of actual neuronal loss in a primate injury model must work within the guidelines of ethical animal research. Therefore, we have triangulated evidence from three distinct sources that indicate the electrolytic lesioning device reliably delivered current that caused tissue damage and neuron death (termination): ex vivo and in vivo histology (*Figures 2 and 3*), lesion device readouts (*Figure 6*), and relative changes in the turnover rate of recorded neurons in vivo (*Figure 7C and D*, case 1).

Temporary increases in the day-to-day turnover of recorded waveforms were observed after lesioning (i.e. decreases in the percent of matching neurons). In addition to increases in turnover characteristic of pre-lesion data, we further identified statistically new clusters of lesion evoked perturbations to low-dimensional projections of recorded neurons' waveforms. On one hand, this is likely an underestimate, as significant neuronal loss is likely occurring just outside of the array electrodes' recording volume and thus evading detection. However, there is also the potential to overestimate loss as well, given it is impossible to disambiguate between neurons that were silenced but surviving versus truly terminated. We note that these sources of error may be somewhat counterbalanced. The evidence provided here suggests a temporary increase in the recorded waveforms' turnover after a lesion, which likely captures evoked neuron loss in addition to lesion-evoked changes in the proportions of activity recorded among neurons.

It is evident in our recordings that many of the array's electrodes are recording smaller inseparable waveforms (hash) corresponding to many different neurons located between between 50 μm and 140 μm from the electrode tip before the lesion (*Buzsáki, 2004*; *Gerstein and Clark, 1964*). Even if the closest, cleanly separable neurons were terminated by the lesion, the recorded signal would still be comprised of the remaining, surviving neurons in that vicinity. These neurons may also become more active after a lesion, resulting in similar estimated firing rates. This was likely the case for the first lesion in Monkey U (*Figure 7A*, top) and lesion six (Monkey H; *Figure 7A* bottom), where the average waveforms on the two lesion electrodes themselves appear similar with only slight differences in amplitude.

While we have thoroughly tested this electrolytic lesioning device to ensure proper function, we emphasize that this is an initial design of the device and a preliminary examination of lesion parameters. We have only begun to explore the full potential for electrolytic lesioning through a microelectrode array, in both its technical implementation and its future use in systems neuroscience. This technique was tested across three species, before settling on a combination of current amplitude and duration that seemed reasonable for experiments in awake-behaving rhesus macaques. Thorough testing had to be balanced with care and respect for appropriate animal use in accordance with the 3 R's of the guidelines of the *National Research Council, 2003*; *National Research Council, 2011*. Future alterations could be made to the design of the lesioning device itself to optimize lesioning through this platform. For example, electronic control of current delivery for lesioning could be refined through the use of digital switching and a resistor shunt to handle any leakage current. Even with such automation, it would be prudent to maintain a physical switch to prevent extremely long periods of current delivery in the case of a malfunction. A circuit revision with these refinements has been designed and validated, but it has not yet been tested in animals out of aforementioned respect for animal use.

Further exploration into the amplitude and duration parameters of the current used to lesion could lead to markedly different impact on brain tissue. Although these lesions were created with direct current, a variety of current patterns could be used to create a lesion. Catheter ablation of the heart used to be performed with direct current (*Scheinman et al., 1982*; *Gallagher et al., 1982*). Now, the standard of care is using a radiofrequency (350–500 kHz) alternating current (*Shivkumar, 2019*; *Morady, 1999*). Similarly, some have explored using radiofrequency (55 kHz) alternating current to create lesions to mark the location of acute electrophysiological recordings (*Brozoski et al., 2006*). It is likely that alternating current or more complex current patterns would lead to different spatial distributions of electrolytic lesions. The PA97 op-amp used in this lesioning device has a slew rate of $8V/\mu s$, and can support AC frequencies in the tens to hundreds of kHz range for the lesioning voltages seen here. At the same time, alternating currents would likely introduce new reactive parasitics, potentially along the electrode wire bundle or at interconnects, and should be evaluated carefully.

Different selection of electrodes within the microelectrode array could also lead to different lesion characteristics. We chose to lesion with adjacent electrodes only, in order to create the most focused lesion. In theory, larger ablations are possible by supplying larger currents or performing a lesion over non-adjacent electrodes, which can span as much as 5.6 mm for electrodes on the opposite corners of the 4 mm x 4 mm Utah array.

Electrode shape could also be used to create different lesions. In deep brain stimulation treatment for Parkinson's disease, new electrodes were designed to directionally focus the stimulation (*Steigerwald et al., 2019*). Similar design could be used in creating intracortical electrodes optimized for both elctrophysiology and electrolytic lesioning. We demonstrated that our electrolytic lesioning technique works with a linear multicontact probe by testing with a U-Probe in ex vivo rabbit cortex. There are no particular limitations that would prevent our specific electrolytic lesioning technique and device from working with any passive multielectrode probe. The main requirements for use are that the probe has two electrodes that can directly (via whatever necessary adapters) connect to the lesioning device, such that arbitrary current can be passed into them as the anode and cathode. This would limit use of probes, like Neuropixels, where the on-chip acquisition and digitization circuitry generally precludes direct connection to electrodes (*Jun et al., 2017*; *Steinmetz et al., 2021*). The impedance of the multielectrode probe should not be an issue, due to the use of an op amp. We showed use with a Utah array (20–800 kΩ) and a U-Probe (1–1.5 MΩ). The specific op amp used here has a voltage range of ±450 V, which assuming a desired output of 150 μA of current would limit electrode impedance to 6 MΩ. Although a different op amp could easily be used to accommodate a higher electrode impedance, it is unlikely that this would be necessary, since most electrodes have impedances between 100 kΩ to 1 MΩ (*Maynard et al., 1997*).

Perhaps the most unique advantage of our technique in comparison with other existing inactivation methods lies in Design Consideration #1: stable electrophysiology pre- and post-inactivation (Appendix 1). While several methods exist that allow for localization and size control of the inactivation (Design Consideration #2) and cross compatibility across regions and species (Design Consideration #3), few have achieved compatibility with stable electrophysiology. For example, some studies record electrophysiology only after the creation of the lesion, preventing comparison with baseline neuronal activity (*Khanna et al., 2021*). One recent study, (*Khateeb et al., 2022*), developed an inactivation method that is effectively combined with stable electrophysiology by creating photothrombotic lesions through a chronic cranial window integrated with an electrocorticography (ECoG) array (*Khateeb et al., 2022*), which may be appropriate for applications where local field potential (LFP) recording is sufficient. This approach has trade-offs with regards to the three design considerations presented in Appendix 1.

While Khateeb et al., present a toolbox with integrated, stable electrophysiology from an ECoG array pre- and post- inactivation (Design Consideration #1), it demonstrated recordings from an ECoG array with limited spatial resolution. While a higher density ECoG array that would provide higher spatial resolution could be used, increasing the density of opaque electrodes might occlude optical penetration and constrain photothrombotic lesions. Further, ECoG arrays are limited to recording LFP, not electrophysiology at single neuron resolution, potentially missing meaningful changes in the neuronal population activity after lesioning. Khateeb et al., demonstrated localization and control the size of inactivation (Design Consideration #2). In this manuscript, we have shown that the amount and duration of direct current are significant determinants of lesion size and shape, while with photothrombotic lesions, light intensity and aperture diameter are the significantly relevant parameters. One potential advantage of photothrombotic approaches is the use of optical tools to monitor anatomical and physiological changes after lesioning through the cranial window, though the research utility of this monitoring remains to be demonstrated.

Although the method presented by Khateeb et al., shows some cross-compatibility (Design Consideration #3), it has greater limitations in comparison with the method presented here. For example, while Khateeb et al., notes that the approach could be adapted for use in smaller organisms, no modification is needed for use in other species with this work's approach–so long as a multielectrode probe is implantable. In this manuscript we demonstrate electrolytic lesioning spanning two multielectrode probes across rabbits, pigs, sheep, and rhesus macaques, and our same device could be easily used with other smaller species, like rats, in which multielectrode probes have been successfully implanted (*Black et al., 2018*). Further, the approach in Khateeb et al., is limited to superficial brain structures,

due to the need for optical accessibility. As noted, fiber optics could allow access to deeper structures, which would bring associated additional tissue damage, but deeper structure lesioning was not demonstrated. In contrast, the approach presented here can be used in any region of cortex in which a multielectrode probe can be implanted, which, depending on the probe used, does not limit it to surface structures. For example, we demonstrated use of our lesioning technique with a linear U-probe (*Figure 8*), which could be used to reach deeper layers of cortex or specific deep cortical structures. In both techniques, the location of the lesion is tied to the location of the electrophysiology (for Khateeb et al., wherever the cranial window and ECoG array are; for this technique, wherever the multielectrode probe has been implanted), which ensures that the electrophysiology will include recordings from the perilesional area. Neither work addresses the potential of their technique to induce chronic post-lesion behavioral effects, which is a key goal for future work.

The electrolytic lesioning method presented here enables small, permanent inactivation volumes while maintaining reliable neuroelectrophysiological recordings. Selection of the electrical current pattern, amplitude, and duration, as well as the specific lesion electrodes' shape and location, offers many different combinations for complementary investigations into the causal role of cortical activity. Further, the same microelectrode array used to create the lesion and record electrophysiology can be used for intracortical microstimulation. This additional aspect of our platform will allow for testing of intracortical microstimulation as a possible treatment or therapy after neuronal loss.

## Materials and methods
### Electrolytic lesioning device design

In order to control lesion size, a current source is required to stabilize the output, due to changes in local tissue resistivity and the design of intracortical electrodes. Single unit electrodes are generally coated with some type of dielectric material to maintain a low surface area of exposed metal, yielding a high impedance. This high impedance enables the detection of the weak currents (order tens of nA) associated with single neuron action potentials with voltages on the scale of tens to hundreds of microvolts (*Carter and Shieh, 2015*). The shafts of the Utah electrode array are insulated with parylene-C (Blackrock Neurotech, Salt Lake City, UT). A small area at the tip of each electrode remains uncoated, through which the ionic current from nearby neurons can be measured. During electrolytic lesioning, applied voltages can be large enough to etch off the dielectric coating from the shaft of the two electrodes used to lesion. As this coating is etched, the impedance of those electrodes falls. Using a constant voltage source to lesion could deliver an inappropriately large amount of current into the brain once the dielectric coating was etched off and impedance reduced, resulting in uncontrolled tissue damage. A constant current source is robust to this changing impedance, maintaining the desired electrical current.

Single unit electrodes are commonly manufactured to have an impedance on the order of 100 kΩ to 1 MΩ (*Maynard et al., 1997*). In past electrolytic lesioning studies, the currents used to create lesions with corresponding behavioral deficits were on the order of 100 μA (*Glees and Cole, 1950*; *Wurtz and Goldberg, 1972*). Therefore, in order to supply this current in the face of the high impedance dielectric coating associated with single unit electrodes, high tens to low hundreds of volts are needed, depending on the lesioning parameters chosen. Even though commercial devices exist for electrolytic lesioning (e.g. Ugo Basile, Gemonio, IT), the power supplied may not be sufficient to recreate some current amplitudes used to lesion in literature (*Wurtz and Goldberg, 1972*) and generally only support DC waveforms.

We designed and built a custom current source circuit (*Figure 1*) that supplies this voltage while maintaining precision in the delivered current (precision error not exceeding 10 μA). The circuit features a commercial power operational amplifier, PA97 (Apex Microtechnology, Tucson, AZ), in a negative feedback loop *DePaola, 2020*, implemented in accordance with the corresponding evaluation kit, EK28 (Apex Microtechnology, Tucson, AZ). Power is provided through a 12 V external power supply, and a boost converter is used to create a $V_{CC}$ and -$V_{CC}$ of 450 V and –450 V, respectively. These are both low-cost, off-the-shelf, readily-available discrete circuits. The amplitude of the current supplied through between the cathode and anode to the brain (shown as the load, $R_L$) is tuned through altering the value of a variable potentiometer ($R_S$). The voltage supplied to the op-amp can also be altered by altering the bias resistance with the potentiometer, $R_B$. The total cost of the parts

needed is approximately three hundred US dollars, making it an affordable addition to a neuroelectrophysiology recording setup.

## Experimental setup

The lesioning device is built as a contained box, to which the power supply, measurement devices, and microelectrode array are externally connected (*Figure 1—figure supplement 1*). An external switch on the lesioning device controls the power to the boost converter. This is the switch that is manually turned on and off to control the duration of current delivery for lesioning. The lesioning device is connected to two electrodes from the intracortically implanted microelectrode array ($R_L$). All lesioning was performed using Utah electrode arrays with the same specifications (Blackrock Neurotech, Salt Lake City, UT). The arrays are 4 mmx4 mm, with 96 channels. Electrode shafts are made of silicon (Si) with a metallic outer layer (platinum or platinum-iridium), and coated with parylene-C. Electrodes are 1 mm in length and have a 400 µm inter-electrode spacing. The array is connected to the lesioning device with the array's external CerePort (Blackrock Neurotech, Salt Lake City, UT) and a CerePort breakout adaptor that connects to specific electrodes from the array. A Fluke 179 True-RMS Digital Multimeter (Fluke Corporation, Everett, WA) is used as an ammeter in series before $R_L$ to measure the current passing through $R_L$ (the desired output). A voltmeter spans $R_L$ to measure the voltage delivered. The variable potentiometer $R_S$ is tuned by a dial on the outside of the lesioning device, allowing quick and easy calibration of the amplitude of the current.

All animal procedures and protocols were approved by the Stanford University Institutional Animal Care and Use Committee (#D16-00134).

### Ex vivo ovine and porcine testing

Initial testing of lesion parameters was performed ex vivo in three sheep and eight pig brains (*Supplementary file 1*). These un-fixed brains were from animals that were euthanized earlier that day for unrelated research and/or teaching efforts, in line with the reduction principle of animal research *National Research Council, 2003*; *National Research Council, 2011*. As these brains were from unrelated research efforts, we were not given the sex or exact age of the animals. The brain was kept moist with saline throughout the procedure. A Utah array was implanted with a pneumatic inserter (Blackrock Neurotech, Salt Lake City, UT) into suitably large and flat gyri. The lesioning device was then connected to the microelectrode array and used to create a lesion. The location of the array implantation was marked using a histology pen or ink injection and detailed for the research record. The brains were then fixed, sliced, and prepared with hematoxylin and eosin (H&E) staining.

### In vivo anesthetized porcine testing

We next sought to determine how lesion size might be altered in vivo, where blood flow, microhemorrhage, and the body's acute inflammatory responses could affect the lesion extent and intensity over time. Testing was performed on five anesthetized pigs, to evaluate the ex vivo current amplitude and duration parameters in the context of inflammatory responses (*Supplementary file 2*).

Each pig was sedated and then intubated, ventilated, and placed on inhaled isolflurane. Once under anesthesia and the airway was confirmed, the head was secured in preparation for the procedure. After preparing and cleaning the surgical area, a midline skin incision was made and all skin, muscle, aponeurosis, and periosteum were retracted to expose the skull. Most of the dorsal surface of the skull was exposed along with some of the lateral margins to visualize the anatomy. A single, large craniectomy, exposing most of the superior surface of the dura, was made using a high speed bone drill (ANSPACH, DePuy Synthes, Raynham, MA). Once the craniectomy was at the desired size and location, the dura was washed with saline and hemisphere-wide dural flaps were made to expose the brain. Once exposed, the brain was covered with gauze and kept moist with saline throughout the procedure.

The implantation and lesion procedures were completed in the same manner as in the ex vivo testing. After lesioning, the array was either removed by hand and placed at a new cortical location for further testing or, if the last lesion of the procedure, was left in place for two to three hours before euthanizing the animal to even better mimic the post-lesion inflammatory response. Therefore, histology for all lesions included the physiological response to lesioning after at least two to three hours, and some lesions also retained the array in place as a foreign body during this post-lesion time

frame. After the final lesioning was complete and sufficient time had elapsed for lesions to appear histologically, the pig was euthanized via IV injection of beauthanasia solution (100 mg/kg, IV). The brain was then fixed, and slices were prepared with H&E staining.

### In vivo lesions in rhesus macaque

Electrolytic lesioning is not painful as there are no direct pain receptors in the central nervous system (*Hall and Hall, 2021*). Based on this, and a lack of physiological signs of pain from the anaesthetized pig studies, a lesion was performed on a sedated rhesus macaque who was subsequently euthanized due to unrelated health complications (Monkey F; 16-year-old adult, male rhesus macaque) in order to further verify safety before use in awake-behaving rhesus. This lesion was created by applying 150 µA of direct current to two adjacent electrodes in the microelectrode array for 45 s. Again, no physiological signs of stress or complications were observed, increasing confidence for lesions in awake animals. Subsequently, lesions were safely performed in two other rhesus (Monkeys H and U; 14 and 11 years-old, respectively, adult, male rhesus macaques) while awake and seated in a primate chair, without the animals exhibiting any behavioral signs of stress or pain. Across these two animals, fourteen lesions were performed using thirteen unique electrode pairs, demonstrating that the same microelectrode array can be used for several lesions and even the same electrode pair can be used as the anode and cathode for multiple lesions. In these lesions, 150 µA of direct current was applied to two adjacent electrodes in the microelectrode array for 30 or 45 s (30 s for Monkey U, 45 s for Monkey H), except in lesion H200120 where current was applied for 50 s. The Utah array chronically implanted in the monkey's cortex is connected to the lesioning device via its CerePort (the same one used for recording the neuronal signals from the microelectrode array). The lesioning device is then turned on for a fixed duration, delivering the desired current. The animal is continuously monitored during the procedure for any signs of discomfort. After lesioning is complete and the recording cables are re-connected to the CerePort, the monkey is immediately ready to resume electrophysiological recordings and participate in a wide variety of head-fixed or freely moving behavioral paradigms. Although lesioning itself is painless, the technique is intended to cause a temporary functional impairment. In light of this, monitoring of animal behavior post-procedure is conducted by research staff in coordination with veterinary staff to ensure health, safety, and psychological well-being.

## Estimation of lesion volume

In order to explore the relationship between the amplitude and duration of the applied current and lesion volume, we needed to have estimates of lesion volume. Gathering multiple histological slices per lesion would have been cost prohibitive, so we attempted to take a histology slice from the estimated center of the lesion. If the histology slice successfully contained the lesion, we assumed the slice bisected the lesion volume. Occasionally, however, the histology slice did not contain the lesion. For these lesions, if we did have a macro camera photo and the lesion was large enough to create a visible cavitation, we used this photo to estimate the diameter of the lesion, and as no information was available about the height of the lesion cavitation, we estimated the height to be equivalent to the radius. We used the radius and height (gathered either from histology or a photo) to estimate the volume of each lesion, using volume formulas for the approximate geometries observed visually in the histology. Most volumes were estimated assuming a conical lesion volume($V = \pi r^2 \frac{h}{3}$), which was chosen based on *Figure 2* and *Figure 3*. Only one in vivo lesion, where 150 µA direct current was applied for 30 s, was calculated as a spherical volume ($V = \frac{4}{3}\pi r^3$) as indicated by the visually identified damage in the histology slice. When we plotted just lesions created with a direct current amplitude of 180 µA, we fit the data with an exponential curve (curve_fit, SciPy optimize).

## Neuronal recordings and processing

Neuronal data was collected at 30 kHz from primary motor cortex of Monkeys H and U using a Utah array (described above), captured using a Cerebus system (Blackrock Neurotech, Salt Lake City, UT), then high-pass filtered offline with a fourth order, zero-phase Butterworth filter using a 250 Hz cutoff frequency. Action potentials were identified by thresholding the filtered membrane potential at −4× the root mean square voltage measured over the first minute of the recording session. Data snippets for comparing electrophysiology pre- and post-lesion were generated by taking the preceding 16 sampled data points (0.53ms), and proceeding 32 sampled points (1.07ms) around the

peak depolarization identified after each threshold crossing. Waveforms whose peak-to-trough amplitude exceeded 300 μV are atypical of extracellular recordings from cortical neurons using Utah arrays and were excluded (*Chestek et al., 2011*). Waveform width was calculated as the difference between the two prominent peaks in the temporal derivative of the extracellular potential. Unphysiological waveforms whose widths were longer than 1ms were also excluded (*Bean, 2007*). The remaining identified action potential waveforms were then averaged for each electrode, and event rates were determined as the number of spikes on a given electrode divided by the duration of the recording session (excluding brief periods of inactivity before the start of the session).

As a putative indication of neuronal loss, neuron turnover was measured between daily recording sessions (each pair separated by at most three days, a time frame that minimizes natural turnover in recorded waveforms *Gallego et al., 2020*). For a given electrode, 1000 identified action potential samples each 1.6ms, 48 points sampled at 30 kHz from the first day's recording session were peak-aligned to create a 1000x48 matrix, where each row contains the 48 time points of a given waveform sampled at 30 kHz. Each of these points in time were considered as a sample feature and principal component analysis was performed. Next, the second day's waveforms were projected onto the top two principal components of the first day (see *Figure 7B*). The distance of each projected waveform from the origin was calculated for each day; a median radius value was computed and represented as a circle around the origin. Changes in the radius of these circles, $\Delta r$, captures changes in the composition of multi-unit activity recorded at each electrode. By fitting Gaussian mixture models to the distributions of $\Delta r$, we were able to find evidence of discrete clusters of radial changes. One and two standard deviations from the median cluster value were used to define five categories: matching waveforms, small increase, small decrease, large increase, and large decrease. A given electrode's waveforms were said to match if $\Delta r \approx 0$ and was considered altered otherwise. Electrodes with matching waveforms are presented as a percentage of the array's total 96 electrodes for a given pair of days (% match).

These pairwise comparisons were then split up into three groups: pre-lesion versus pre-lesion days (pre-pre), pre-lesion versus post-lesion days (pre-post), and post-lesion version post-lesion days (post-post). Although a set of eleven consecutive days was analyzed for each lesion (four pre- and seven post-), only recording session pairs that were separated by three-days or fewer were analyzed in order to minimize the potential confound of increased neuron turnover resulting from longer separation between sessions (*Gallego et al., 2020*). Comparisons were then made between the pre-lesion period, an acute injury period (up to 3 days following a lesion), and a late injury period (days 4 to 7 following a lesion). The Median test was used to test the null hypothesis that lesioning had no effect on recorded turnover with 95% confidence interval, whose significance threshold was Bonferroni corrected for the three statistical comparisons made between groups ($\alpha/3$, significant if $p < 0.017$).

## Ex vivo testing with a linear multielectrode probe

Testing of the ability to lesion through a different type of multielectrode probe (a linear multielectrode probe, rather than the microelectrode arrays used throughout the rest of the study) was done in an ex vivo rabbit brain. The un-fixed brain was from an animal that was euthanized earlier that day for unrelated research efforts. As this brain was from an unrelated research effort, we were not given the sex or age of the animal. The brain was kept moist with saline throughout the procedure. A U-Probe (Plexon, Dallas, TX), with 24 contacts, 100 μm spacing between them, and an impedance of 1–1.5 MΩ, was lowered one cm into cortex using a stereotaxic manipulator. Using a modified connector that attached to the female connector at the top of the probe, we connected our electrolytic lesioning device to two of the contacts on the probe, to act as the anode and cathode through which current was passed. The two contacts were contiguous (100 μm apart), and the lesions were created at depths approximately 400 μm below the surface of cortex. The location of the probe insertion was marked using ink injection, and is indicated in a sketch in *Figure 8A*.

## Acknowledgements

We thank S Baker for veterinary support, and M Truong and K Chin for administrative support. The members of the Brain Interfacing Laboratory are Kristina Lebedev, Michelle S Wechsler, Mackenzie J Risch, Stephen I Ryu, Alissa S Ling, Michael P Silvernagel. MS Wechsler and MJ Risch were responsible for animal care, surgical support, and guiding behavioral training. SI Ryu was responsible for

nonhuman primate array implantation. AS Ling and MP Silvernagel assisted in animal care and porcine surgeries. Research reported in this publication was supported by the Stanford University Wu Tsai Neurosciences Institute, an American Heart Association Predoctoral Fellowship - 828653 to IEB, a NSF GRFP DGE - 1656518 to IEB, and a Stanford School of Medicine's Dean's Posdoctoral Fellowship to SEC.

## Additional information

### Funding

| Funder | Grant reference number | Author |
|---|---|---|
| American Heart Association | Predoctoral Fellowship - 828653 | Iliana E Bray |
| National Science Foundation | Graduate Student Research Fellowship - 1656518 | Iliana E Bray |
| Stanford School of Medicine | Dean's Posdoctoral Fellowship | Stephen E Clarke |
| Stanford Wu Tsai Neurosceinces Institute | | Paul Nuyujukian |

The funders had no role in study design, data collection and interpretation, or the decision to submit the work for publication.

### Author contributions

Iliana E Bray, Formal analysis, Validation, Investigation, Visualization, Methodology, Writing - original draft, Writing – review and editing; Stephen E Clarke, Formal analysis, Validation, Investigation, Visualization, Methodology, Writing – review and editing; Kerriann M Casey, Data curation, Writing – review and editing; Paul Nuyujukian, Conceptualization, Resources, Supervision, Funding acquisition, Validation, Investigation, Methodology, Project administration, Writing – review and editing

### Author ORCIDs

Iliana E Bray ![ORCID] https://orcid.org/0000-0002-3029-8309
Stephen E Clarke ![ORCID] http://orcid.org/0000-0002-3871-0185
Kerriann M Casey ![ORCID] https://orcid.org/0000-0003-4228-928X
Paul Nuyujukian ![ORCID] https://orcid.org/0000-0001-7778-5473

### Ethics

All animal procedures and protocols were approved by the Stanford University Institutional Animal Care and Use Committee (#D16-00134).

Reviewer #1 (Public Review): https://doi.org/10.7554/eLife.84385.3.sa1
Reviewer #2 (Public Review): https://doi.org/10.7554/eLife.84385.3.sa2
Author response https://doi.org/10.7554/eLife.84385.3.sa3

## Additional files

### Supplementary files

• Supplementary file 1. Lesion parameters used for ex vivo testing. Voltage was monitored with a voltmeter during lesioning, and notes were collected about the voltage. Tests that were performed solely to understand the effect of impacting and removing the microelectrode array without passing any current to create an electrolytic lesion are indicated with N/A for the current value. One ex vivo brain was used for all testing on 180702, and two ex vivo brains were used on each of the other two dates.

• Supplementary file 2. Lesion parameters used for in vivo testing. Voltage was monitored with a voltmeter during lesioning, and notes were collected about the voltage. Tests that were performed

solely to understand the effect of impacting and removing the microelectrode array without passing any current to create an electrolytic lesion are indicated with N/A for the current value. One animal was used for all testing on a given date.

• MDAR checklist

## Data availability

Data not available within figures and tables is available at https://doi.org/10.25936/j6nd-mp50.

The following dataset was generated:

| Author(s) | Year | Dataset title | Dataset URL | Database and Identifier |
|---|---|---|---|---|
| Bray IE, Clarke SE, Casey K, Nuyujukian P | 2024 | Neuroelectrophysiology-Compatible Electrolytic Lesioning | https://doi.org/10.25740/ct335mc2298 | Stanford Digital Repository, 10.25740/ct335mc2298 |

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

# Appendix 1

## Design Considerations

A technique that combines electrophysiology with permanent inactivation of neuronal activity will meet three main design considerations:

1. stable electrophysiology pre- and post-inactivation
2. ability to localize and control the size of the inactivation
3. cross-compatibility

## Stable electrophysiology pre- and post-inactivation

Stable electrophysiological recordings over time allow baseline neuronal activity to be directly compared to the activity recorded after inactivation. This comparison would capture the acute effects of inactivation on both the neuronal activity and the animal's behavior, as well as long term changes associated with behavioral recovery. As any physical disruption of the multielectrode probe would affect the stability of the recordings and negatively impact pre-post comparisons, it is essential to avoid physical disruption of the array from surgical procedures and minimize additional implanted devices or cannulae, which may act as routes for infection to enter the brain.

## Ability to localize and control the size of the inactivation

Prior studies suggest that after inactivating neuronal activity in a region, compensatory neuronal changes will happen in the area closely surrounding that region (*Gould et al., 2021*; *Nudo, 2013*; *Nudo and Friel, 1999*). In order to ensure that the multielectrode probe is recording from this peri-inactivation area, the location of the inactivation (both point of origin and extent) must be precisely controlled. The multielectrode probe should be close enough to the inactivation to record from the peri-inactivation area, but the inactivation area should not be so large or so close to the multielectrode probe that it completely encompasses the array's recording area. If the multielectrode probe only records from the inactivated area (and therefore inactivated neurons), the recordings will only verify that the inactivation method worked and will not enable further scientific questions.

While some methods do allow for inactivation at a certain point in space, the effects can spread quite far or uncontrollably from that point due to various biological mechanisms. These mechanisms can include pharmacological diffusion and continued neuronal death after ischemic injury (*Jarrard, 2002*; *Kubota, 1996*; *Schieber and Poliakov, 1998*). This spread would not allow control over how much of the recording area of the multielectrode probe was in the peri-inactivation region or the inactivation region itself. Therefore, both precision in localizing the inactivation and control over its spread are required.

Finally, an inactivation technique should enable studying both the effects of inactivation on neuronal activity and how neuronal activity changes when behavior recovers following inactivation (*Slonina et al., 2022*). This requires the inactivation to be small enough for the animal to recover (*Glees and Cole, 1950*; *Nudo et al., 2003*; *Wurtz and Goldberg, 1972*). Inactivating a small region may enable investigation about the causal role of the inactivated neurons while avoiding confounding effects across multiple, broad systems.

## Cross-compatibility

Ideally, a technique that combines electrophysiology with inactivation could be used in any area of cortex, with any multielectrode probe, and in several species in order to enable causal investigation in a large variety of contexts. Flexibility over the cortical area studied and the multielectrode probes used will enable easy adoption into existing neuroelectrophysiology recording setups across many neuroscience contexts. Compatibility with a variety of animal models will further increase the technique's utility. Electrophysiology studies and animal models of brain disease have spanned many species, from rodents to large animals (*Fan et al., 2017*; *Finnie and Blumbergs, 2002*; *Le et al., 2014*; *Lind et al., 2007*; *Nudo et al., 2003*; *Rousche and Normann, 1999*; *Rousche and Normann, 1998*; *Kleinschnitz et al., 2015*). While each species offers its own experimental benefits and limitations, the majority of tools and techniques in this space have been developed for rodents. Rather than being limited to rodents, an inactivation technique would ideally also be compatible with large animals such as rhesus macaques, due to their dexterity, ability to perform complex movements, and extensive history in electrophysiology studies of motor control (*Higo, 2021*).

# Appendix 2

Manipulations are temporary inactivations of neuronal activity, while terminations are permanent inactivations. There are several existing methods for achieving a manipulation or a termination. Although each existing method has its own strengths, none is able to meet all three of the previously mentioned design considerations of: stable electrophysiology pre- and post-inactivation, ability to localize and control the size of the inactivation, and cross-species and large-animal compatibility.

## Existing Manipulation Methods

Existing temporary inactivation methods include intracortical microstimulation, optogenetics, pharmacology, transcranial stimulation, cooling loops, and chemogenetics.

Intracortical microstimulation, where small pulses of current are applied to cortex, can be used to temporarily disrupt neuronal activity (*Churchland and Shenoy, 2007*; *Mazurek and Schieber, 2017*; *Vyas et al., 2020*). It can be performed using the same neuroelectrophysiology electrodes being used to record, requiring no additional surgical access (*Weiss et al., 2019*). However, it can be challenging to sustain a behavioral effect with continuous microstimulation.

Optogenetic silencing can inactivate an area either by using an inhibitory step-function opsin (*Berndt et al., 2014*; *Kim et al., 2017*) or by activating a local inhibitory circuit (*Li et al., 2019*; *Vogt, 2020*). Long-term local silencing of a region of cortex could be achieved with continued illumination by a fiber or a chronically implanted array of light-emitting diodes (*Rajalingham et al., 2021*). However, other challenges exist with using optogenetics as an inactivation method in nonhuman primates, including difficulty reliably affecting behavior (*Afraz, 2023*). While several constructs for rhesus macaques have been developed (*Galvan et al., 2017*; *Tremblay et al., 2020*), reports of successfully inducing behavioral effects have a small effect size and are less numerous than might be expected (*Afraz, 2023*), and several null results have been published (*Diester et al., 2011*; *Galvan et al., 2016*; *O'Shea et al., 2022*). Other remaining challenges include the need to develop a head-mounted, battery powered light delivery system for multi-day delivery of light and difficulty integrating illumination with simultaneous chronic neuroelectrophysiology.

Pharmacological agents like muscimol and lidocaine can also be used for transient inactivation (*Clarke and Maler, 2017*; *Kubota, 1996*; *Schieber and Poliakov, 1998*). A pathway is required to deliver the agent to the appropriate area of cortex. If this pathway is chronically implanted (e.g., a cannula), then it can be placed somewhat precisely near the multielectrode probe, but it would act as a potential route for a local infection, which may cause swelling of the tissue and lead to partial displacement of the array or other medical complications. An alternative is to inject the agent though a burr hole created only when lesioning is desired, though it may be difficult to localize the placement of the burr hole within the immediate area of the multielectrode probe. Controlling spread of pharmacological agents is also difficult, especially for drugs with low molecular weight. The effects of pharmacological agents will fluctuate over space and time as they diffuse through the tissue and begin to be cleared or metabolized.

Non-invasive methods like transcranial magnetic stimulation (TMS), transcranial direct stimulation (tDCS), and transcranial focal ultrasound lead to very temporary inactivation (*Klomjai et al., 2015*; *Woods et al., 2016*; *Zhang et al., 2021*). In addition to the short timescale of these inactivations, their effects are dispersed as the signal must travel through the skull. These methods provide neither the spatial resolution nor the temporal duration needed for probing local neuronal circuitry.

Cooling loops are implanted devices that effectively silence nearby cells by affecting action potential generation, axon conduction velocity, and synaptic transmission (*Chen et al., 2020*; *Lomber and Payne, 2000*; *Lomber et al., 1999*; *Long and Fee, 2008*). These devices would be implanted at the same time as the multielectrode probe, and they would not require surgical access to use. However, cooling cannot easily be maintained across days while the animal is in its home environment. In addition to damaging valuable nearby tissue during the implant surgery and introducing another foreign body, cooling inactivates on the scale of millimeters, limiting the ability to titrate the size of the inactivation any smaller (*Coomber et al., 2011*).

A region of cortex can also be temporarily inactivated with chemogenetic silencing, using chemically activated proteins to inhibit neuronal activity (*Sternson and Roth, 2014*). The chemical is injected intravenously, so the process would not disrupt the placement of the multielectrode probe. As with optogenetic silencing, chemogenetics would require the development of rhesus compatible

constructs. Additionally, chronic inactivation over days may be logistically challenging, as the half life of clozapine N-oxide (CNO, a ligand used to activate DREADD receptors) is on the order of hours.

## Existing Termination Methods

Existing termination methods include physical damage, endovascular occlusion, Rose Bengal mediated photothrombosis, and chemical lesioning.

There are several well established techniques for mechanically damaging a small area of cortex to create a lesion, including blade lesioning (*Horsley and Schafer, 1888*; *Sherrington, 1893*), vacuum aspiration (*Darling et al., 2016*), vascular cauterization (*Nudo et al., 2003*), and vascular ligation (*Rumajogee et al., 2016*). All of these techniques require surgical access to cortex, which would likely disrupt an existing implanted microelectrode array. Additionally, the sedation necessary for the surgery would prevent behavioral testing on the day of the lesion, precluding measurements of acute inactivation. Further, these techniques often create large lesions, and do not offer sub-millimeter precision.

Endovascular techniques are commonly used as models of stroke. For example, endovascular physical occlusion of the middle cerebral artery (MCA) of one hemisphere is a common rodent model of stroke (*Kleinschnitz et al., 2015*), but it is challenging to precisely control the extent of cortical damage. MCA occlusion could cause indiscriminate injury to a large region of cortex, due to continued, widespread neuronal death after the occlusion. It could potentially damage the area in which the multielectrode probe is implanted, preventing meaningful recordings. As one descends into smaller branches of the MCA, survivability, localization, and reproducibility of ischemic results improve (*Clark et al., 2019*; *Kuraoka et al., 2009*), but it is technically challenging to be precise with the occlusion without coming close to the implanted array, again risking disrupting the implantation site.

Another endovascular technique is photothrombosis, which does not require an additional surgery to implement, limiting disruption of the multielectrode probe (*Gulati et al., 2015*; *Khateeb et al., 2022*; *Khateeb et al., 2019*; *Ramanathan et al., 2018*). This approach uses rose bengal, a photosensitive dye, injected intravenously into the circulatory system. When 561nm green light is shined over a blood vessel, the dye undergoes a local conformational change and generates singlet oxygen, damaging arterial endothelial cells and initiating the clotting cascade–resulting in damage resembling an ischemic stroke (*Carmichael, 2005*; *Watson et al., 1985*). This approach can be used to deliver a well-localized lesional boundary. In rodents, this can be done entirely non-invasively because green light penetrates through the thin layer of skull. In larger animals, a method of light delivery is needed. If an optical fiber is chronically implanted at the time of electrode array insertion, light can be delivered without surgery and lesions can be made without disrupting the array, but this chronically implanted fiber may act as a route for infection. Alternatively, the fiber could be placed though a burr hole made at the time of the lesion, but this may compromise localization accuracy like other burr-hole techniques. Another light delivery method for photothrombosis, compatible with use in large animals, is through a cranial window (*Khateeb et al., 2022*; *Khateeb et al., 2019*).

Chemical lesioning is done by injecting a damaging chemical into the cortical region. These chemicals can be excitotoxic pharmacologic agents like ibotenic acid that selectively and directly damage neuronal cell bodies *Murata et al., 2008*, or they can be vasoconstrictors like endothelin-1 that create anoxic cortical injury (*Dai et al., 2017*). These chemicals have the same potential drawbacks of other injection-based methods: either a permanent pathway is added to allow precise injection in the area of the multielectrode probe, creating a route for infection, or injection is done through a burr hole, making it difficult to localize to the region of the array and disrupting experimental continuity. It is also difficult to control the spread of the chemicals, preventing precision in lesion extent.

