## [Editor Report · eLife assessment]

This paper reports a **valuable** new method for creating localized damage to candidate brain regions for functional and behavioral studies. The authors present **solid** support for their ability to create long-term local lesions with mm spatial resolution. The paper is likely to be of broad interest to brain researchers working to establish causal links between neural circuits and behavior.

---

## [Referee Report · Reviewer #1 (Public Review)]

In the paper, the authors illustrated a novel method for Electrolytic Lesioning through a microelectronics array. This novel lesioning technique is able to perform long-term micro-scale local lesions with a fine spatial resolution (mm). In addition, it allows a direct comparison of population neural activity patterns before and after the lesions using electrophysiology. This new technique addresses a recent challenge in the field and provides a precious opportunity to study the natural reorganization/recovery at the neuronal population level after long-term lesions. It will help discover new causal insights investigating the neural circuits controlling behavior.

Comments on revised version:

We appreciate the revisions made by the authors in response to our comments on the previous version of their manuscript. They carefully addressed the majority of the concerns and performed additional experiments. The new figure illustrating the lesion volume as a function of electrolytic lesioning parameters provides a valuable reference for future experiments. In addition, the latest results on different versions of passive multielectrode probes, U-probe, demonstrate that the technique is applicable beyond the specific technical setup they employ. Overall, we believe that the revised manuscript is significantly improved.

---

## [Referee Report · Reviewer #2 (Public Review)]

This work by Bray et al. presented a customized way to induce small electrolytic lesions in the brain using chronically implanted intracortical multielectrode arrays. This type of lesioning technique has the benefit of high spatial precision and low surgical complexity while allowing simultaneous electrophysiology recording before, during, and after the lesion induction. The authors have validated this lesioning method with a Utah array, both ex vivo and in vivo using pig models and awake-behaving rhesus macaques. Given its precision in controlling the lesion size, location, and compatibility with multiple animal models and cortical areas, the authors believe this method can be used to study cortical circuits in the presence of targeted neuronal inactivation or injury and to establish causal relationships before behavior and cortical activity.

Strengths:

- Overall the techniques, parameters, and data analysis methods are better described in the revised version.

- The authors added the section "Relationship Between Applied Current and Lesion Volume" as well as Figure 4 and 5 to address our comments regarding parameter testing. Multiple combinations of current amplitude and duration were tested and the induced lesion volumes were estimated, providing a better picture of why certain parameters were chosen for in vivo studies.

- The authors added Figure 7 which addressed our comment "more evidence is needed to suggest robust neuronal inactivation or termination in rhesus macaques after electrolytic lesioning." They went into more details to explain the observed changes in pairwise comparisons of spike waveforms (difference in projected radii). Particularly in Fig 7C, they identified a new cluster from the pre-post lesioning group, which effectively represented neuronal loss from the

recorded population.

- The authors discussed their method in the context of other literature and stating its strength and limitation.

Major comments:

-The lack of histology limits the validation of lesion induction, ideally cell loss and neuronal loss in vivo needs to be quantified. In addition based on the lack of access to histology, it is not clear how the lesion volumes are calculated which also impacts the scientific rigor of the work. The authors mention that layers 2/3 and maybe 4 have been impacted. The lack of information on the extent of the lesion severely limits the use of their technique for neuroscience experiments.

-The lack of histology in combination with behavioral measures still limits the impact of the paper in the context of NHP research.

- Figure 5 involves fitting an exponential model to the generated lesion volume given the applied current amplitude and duration. However, the data from ex vivo sheep and pig cortex with the same current amplitude & three durations showed very large variability in lesion volume at Time = 2min (larger than the difference from 2 to ~2.2min). Very limited data points exist for the other two parameter combinations. These may suggest that the exponential fit is not the best model in this scenario.

- Regarding the comment on neuronal inactivation, the authors still did not show any evidence of single unit activity loss or changes in local field potential/multi-unit activity from the region being lesioned.

- Regarding this comment "The lesioning procedure was performed in Monkey F while sedated, but no data was presented for Monkey F in terms of lesioning parameters, lesion size, recorded electrophysiology, histological, or behavioral outcomes. It is also unclear if Monkey F was in a terminal study" the authors explained that "a lesion was performed on a sedated rhesus macaque (monkey F) who was subsequently euthanized due to unrelated health complications, in order to further verify safety before use in awake-behaving rhesus" but still no histology data is shown regarding monkey F to demonstrate this verification. Given that NHPs are highly valuable resources, it's important to make use of all collected data and to show that the induced lesion is comparable to those in the pig cortex.

---

## [Author Response]

The following is the authors’ response to the current reviews.

We thank the reviewers and editor for their careful review of our work. We believe the resulting manuscript is much stronger. We agree with the comments made by Reviewer #2 regarding additional histology and neuronal data analysis, which will be presented in subsequent work.

The following is the authors’ response to the original reviews.

**Reviewer 1 (Public Weaknesses):**
It was not always clear what the lesion size was. This information is important for future applica- tions, for example, in the visual cortex, where neurons are organized in retinotopy patterns.

We thank the reviewer for this feedback. While there is some variation in lesion volume for a given parameter set, we have added more details of the volumes of lesions created in our testing (Fig. 4 and Fig. 5).

It would be helpful if the author could add some discussion about whether and how this method could be used in other types of array/multi-contact electrodes, such as passive neuropixels, S- probes, and so on. In addition, though an op-amp was used in the design, it would still be helpful if the author could provide a recommended range for the impedance of the electrodes.

We thank the reviewer for this suggestion. We have both added a demonstration of use in a differ- ent multielectrode probe type (with a U-probe) in Fig. 8, and we have added a discussion about which types of multielectrode probes would be suitable on Page 15, Line 420.

“We demonstrated that our electrolytic lesioning technique works with a linear multicontact probe by testing with a U-Probe in ex vivo rabbit cortex. There are no particular limitations that would prevent our specific electrolytic lesioning technique and device from working with any passive multielectrode probe. The main requirements for use are that the probe has two electrodes that can directly (via whatever necessary adapters) connect to the lesioning device, such that arbitrary current can be passed into them as the anode and cathode. This would limit use of probes, like Neuropixels, where the on-chip acquisition and digitization circuitry generally precludes direct connection to electrodes [1], [2]. The impedance of the multielectrode probe should not be an issue, due to the use of an op amp. We showed use with a Utah array (20-800 kΩ) and a U-Probe (1-1.5 MΩ). The specific op amp used here has a voltage range of *±* 450 V, which assuming a desired output of 150 µA of current would limit electrode impedance to 6 MΩ. Though a different op amp could easily be used to accommodate a higher electrode impedance, it is unlikely that this would be necessary, since most electrodes have impedances between 100 kΩ to 1 MΩ [3].”

**Reviewer 2 (Public Weaknesses):**
In many of the figures, it is not clear what is shown and the analysis techniques are not well described.

We thank the reviewer for this feedback. We hope that our edits to both the figures and the text have improved clarity for readers.

The flexibility of lesioning/termination location is limited to the implantation site of the multielec- trode array, and thus less flexible compared to some of the other termination methods outlined in Appendix 2.

We thank the reviewer for this point. You are right that the lesioning location is limited to the multielectrode array’s implantation site, while other methods in Appendix 2 do not require prox- imity of the lesion location and the electrophysiology recording site. However, we believe that the closeness of the lesioning location to the microelectrode array is a strength - guaranteeing record- ings from the perilesional area - even with the small negative of reduced flexibility. Multielectrode arrays can be implanted in many areas of cortex. If one wanted to study distal effects of a lesion, additional electrophysiology probes could be implanted to record from those areas. We have noted this on Page 3, Line 117.

“While the link between the lesion location and the multielectrode location technically con- strains the lesion to an area of cortex in which a multielectrode array could be implanted, we see the connection as a positive, because it ensures recording some neuroelectrophysiology from the perilesional area in which recovery is hypothesized to occur (see Appendix 1).”

Although the extent of the damage created through the Utah array will vary based on anatomical structures, it is unclear what is the range of lesion volumes that can be created with this method, given a parameter set. It was also mentioned that they performed a non-exhaustive parameter search for the applied current amplitude and duration (Table S1/S2) to generate the most suitable lesion size but did not present the resulting lesion sizes from these parameter sets listed. Moreover, there’s a lack of histological data suggesting that the lesion size is precise and repeatable given the same current duration/amplitude, at the same location.

We thank the reviewer for this thoughtful feedback. We have added figures (Figs. 4 and 5), where we show the relationship between estimated lesion volume and the current amplitude and duration parameters. These figures include more data from the tests in Supplementary File 1 and Supplementary File 2. While there is some variation in lesion volume for a given current amplitude and duration, there is still a clear relationship between the parameters and lesion volume.

It is unclear what type of behavioral deficits can result from an electrolytic lesion this size and type (∼3 mm in diameter) in rhesus macaques, as the extent of the neuronal loss within the damaged parenchyma can be different from past lesioning studies.

While we appreciate the reviewer’s interest in the behavioral deficits associated with our lesions in rhesus macaques, reporting these falls beyond the scope of this manuscript. Future work will explore the behavioral deficits associated with these lesions

The lesioning procedure was performed in Monkey F while sedated, but no data was presented for Monkey F in terms of lesioning parameters, lesion size, recorded electrophysiology, histological, or behavioral outcomes. It is also unclear if Monkey F was in a terminal study.

We apologize for not being more explicit about the parameters used for the lesion in Monkey F. We have added this in Results on Page 5, Line 209 and in Methods on Page 19, Line 586.

“After this validation and refinement, one proof-of-concept lesion (150 µA direct current passed through adjacent electrodes for 45 seconds) was performed in an in vivo sedated rhe- sus macaque (Monkey F) in order to validate the safety of the procedure.”

“This lesion was created by applying 150 µA of direct current to two adjacent electrodes in the microelectrode array for 45 seconds.”

We also clarified the parameters used for the other lesions in Monkeys H and U in Results on Page 7, Line 233 and in Methods on Page 19, Line 586.

“In all of the fourteen lesions across two awake-behaving rhesus macaques (150 µA direct current passed through adjacent electrodes for 30 or 45 seconds (30s for Monkey U and 45s for Monkey H, except lesion H200120 which was for 50 seconds)), the current source worked as expected, providing a constant current throughout the duration of the procedure.”

“In these lesions, 150 µA of direct current was applied to two adjacent electrodes in the mi- croelectrode array for 30 or 45 seconds (30s for Monkey U, 45s for Monkey H), except in lesion H200120 where current was applied for 50 seconds.”

Monkey F was euthanized shortly after the lesion, so we now mention this on Page 19, Line 583.

“Based on this, and a lack of physiological signs of pain from the anaesthetized pig studies, a lesion was performed on a sedated rhesus macaque who was subsequently euthanized due to unrelated health complications (Monkey F; 16 year-old adult, male rhesus macaque) in order to further verify safety before use in awake-behaving rhesus.”

Because Monkey F was sedated and then euthanized shortly after, there is no behavioral data. As the lesion in sedated Monkey F was used to validate the safety of the procedure, any further data and analysis fall beyond the scope of this manuscript.

As an inactivation method, the electrophysiology recording in Figure 5 only showed a change in pairwise comparisons of clustered action potential waveforms at each electrode (%match) but not a direct measure of neuronal pre and post-lesioning. More evidence is needed to suggest robust neuronal inactivation or termination in rhesus macaques after electrolytic lesioning. Some exam- ples of this can be showing the number of spike clusters identified each day, as well as analyzing local field potential and multi-unit activity.

The reviewer has pointed out some short comings of the original analysis, which we believe have since been addressed with the revised analysis. LFP and spiking activity are functional measures that are more ambiguous in terms of loss and are also the subject of another manuscript currently under revision.

The advantages over recently developed lesioning techniques are not clear and are not discussed.

We thank the reviewer for noting this. We have added a section, also responding to their later request for us to compare our work to Khateeb et al. 2022, by adding a section to the Discussion on Page 16, Line 434.

“Perhaps the most unique advantage of our technique in comparison with other existing inactivation methods lies in Design Consideration #1: stable electrophysiology pre- and post-inactivation (Appendix 1). While several methods exist that allow for localization and size control of the inactivation (Design Consideration #2) and cross compatibility across regions and species (Design Consideration #3), few have achieved compatibility with stable electrophysiology. For example, some studies record electrophysiology only after the creation of the lesion, preventing comparison with baseline neuronal activity [4]. One recent study, Khateeb, et al., 2022, developed an inactivation method that is effectively combined with stable electrophysiology by creating photothrombotic lesions through a chronic cranial window integrated with an electrocorticography (ECoG) array [5], which may be appropriate for applications where local field potential (LFP) recording is sufficient. This approach has trade-offs with regards to the three design considerations presented in Appendix 1.

While Khateeb, et al., present a toolbox with integrated, stable electrophysiology from an ECoG array pre- and post- inactivation (Design Consideration #1), it demonstrated recordings from an ECoG array with limited spatial resolution. While a higher density ECoG array that would provide higher spatial resolution could be used, increasing the density of opaque electrodes might occlude optical penetration and constrain photothrombotic lesions. Further, ECoG arrays are limited to recording LFP, not electrophysiology at single neuron resolution, potentially missing meaningful changes in the neuronal population activity after lesioning. Khateeb, et al., demonstrated localization and control the size of inactivation (Design Consideration #2). In this manuscript, we have shown that the amount and duration of direct current are significant determinants of lesion size and shape, while with photothrombotic lesions, light intensity and aperture diameter are the significantly relevant parameters. One potential advantage of photothrombotic approaches is the use of optical tools to monitor anatomical and physiological changes after lesioning through the cranial window, though the research utility of this monitoring remains to be demonstrated.

Although the method presented by Khateeb, et al., shows some cross-compatibility (Design Consideration #3), it has greater limitations in comparison with the method presented here. For example, while Khateeb, et al., notes that the approach could be adapted for use in smaller organisms, no modification is needed for use in other species with this work’s approach–so long as a multielectrode probe is implantable. In this manuscript we demon- strate electrolytic lesioning spanning two multielectrode probes across rabbits, pigs, sheep, and rhesus macaques, and our same device could be easily used with other smaller species, like rats, in which multielectrode probes have been successfully implanted [6]. Further, the approach in Khateeb, et al., is limited to superficial brain structures, due to the need for opti- cal accessibility. As noted, fiber optics could allow access to deeper structures, which would bring associated additional tissue damage, but deeper structure lesioning was not demon- strated. In contrast, the approach presented here can be used in any region of cortex in which a multielectrode probe can be implanted, which, depending on the probe used, does not limit it to surface structures. For example, we demonstrated use of our lesioning tech- nique with a linear U-probe (Fig. 8), which could be used to reach deeper layers of cortex or specific deep cortical structures. In both techniques, the location of the lesion is tied to the location of the electrophysiology (for Khateeb et al., wherever the cra- nial window and ECoG array are; for this technique, wherever the multielectrode probe has been implanted), which ensures that the electrophysiology will include recordings from the perilesional area. Neither work addresses the potential of their technique to induce chronic post-lesion behavioral effects, which is a key goal for future work.”

There is a lack of quantitative histological analysis of the change in neuronal morphology and loss.

We appreciate the reviewer’s desire for a quantitative histological analysis, however this falls out- side of the scope of this manuscript. We are not attempting to make strong claims about the number of neurons lost through lesioning or thoroughly characterize morphological changes in the neurons. The histology is intended to show that lesioning did lead to a loss of neurons, but the precise num- ber of neurons lost is neither in scope nor is likely to be highly conserved across lesions.

There is a lack of histology data across animals and on the reliability of their lesioning techniques across animals and experiments.

We thank the reviewer for this point. As stated above, we have now added Fig. 4 and Fig. 5, which includes volume estimates based on the histology from more of our ex vivo and in vivo testing across animals.

There is a lack of data on changes in cortical layers and structures across the lesioning and non- lesioning electrodes.

We acknowledge that the histology does not have the level of detail that is expected from many modern studies. However, the goal here was dramatically different: we sought to calibrate a novel lesion device, ensure it’s safe use in large mammals (specifically, non-human primates) and pro- vide estimates of the lesion size to compare with the literature. The extent of histology that could be performed and the tools available to us prevent such an in depth analysis. We can say based on shank length of the Utah arrays used and known anatomy that we have affected layer 2/3 and maybe a bit of layer 4.

**Reviewer 1 (Recommendations For The Authors):**
Figure 5b. It would be helpful if the author could plot the delta match separately for the lesion elec- trodes, near neighbor electrodes, and far neighbors. This would help understand the lesion effect, specifically whether the effect is selective (e.g., more potent for the lesion and adjacent electrodes.)

The fact that neuron loss is not particularly selective can already be seen in the spike waveform plots, arranged spatially on the array. Plenty of clear change is observed far from the lesion elec- trodes (marked with black dots) as well as nearby. We have made mention of this localized non- specificity in the main text and have ensured to remphasize in the figure legened. While a nice suggestion, we currently don’t feel this result rises to the level of a figure given it is not highly specific spatially.

**Reviewer 2 (Recommendations For The Authors):**
Overall the quality of the paper, the figures and the analysis used could be significantly improved. There is a lack of scientific rigor in the presentation of figures and analysis techniques. It is not clear what the authors are trying to communicate through the figures and their choice of figures to show is confusing (see below).

We thank the reviewer for their pointed critiques and believe we have addressed their concerns with many changes to the text, a revamped waveforms analysis, and both the expansion and addition of results.

The neurophysiology data shown doesn’t suggest neuronal loss, it only shows change which needs strong control data to show it is due to a lesion.

As detailed below, we have presented a revised analysis that provides this control. While the reviewer is right to point out we can distinguish actual neuron loss from neuron silencing, we be- lieve the new analysis rigorously indicates new rates of sample turnover beyond those expected from healthy state.

The histology figure should be replaced with a high-quality representation without folds.

We understand the reviewer’s suggestion. While ideally we would have many histology slices from each lesion, due to cost, we were only able to collect one histology slice per lesion. The folds were introduced by the company that performed the H&E staining, and we unfortunately cannot remove the folds. Therefore, despite the folds, this is the best and only image from this lesion. We hope that the markings on the figure and the comment in the caption is sufficient to explain to readers that the folds are not a result of the lesion but instead a result of the histology process.

The authors suggest that this lesioning method will be compatible with any available multielec- trode probe theoretically. Since all testing was done with a Utah array, it will be helpful to add an explanation about potential constraints that will make a given array compatible with this method.

We thank the reviewer for this suggestion. As stated above, we have both added a demonstration of use in a different multielectrode probe type (with a U-probe) in Fig. 8, and we have added a discussion about which types of multielectrode probes would be suitable on Page 15, Line 420.

The authors should cite and discuss previous studies using electrolytic lesioning in awake-behaving animals to study the causal connection between the brain and behavior. (One example study: Morissette MC, Boye SM. Electrolytic lesions of the habenula attenuate brain stimulation reward. Behavioural brain research. 2008 Feb 11;187(1):17-26.)

We thank the reviewers for this suggestion. We have added a mention of existing electrolytic le- sioning studies on Page 2, Line 88.

“Prior termination studies mostly measure behavioral output, with no simultaneous measures of neuronal activity during the behavior, impairing their ability to provide insight into the causal connection between the brain and behavior [7]–[11], or with no baseline (i.e., pre- lesion) measures of neuronal activity [4].”

The authors should compare their technique with other recent lesioning studies in primates (e.g. Khateeb et al, 2022)

We again thank the reviewer for this point. Specifically not mentioning Khateeb et al. 2022 was a submission error on our part; we cited the paper in Appendix 2 in the version uploaded to the eLife submission portal, but we had uploaded the version prior to citing it to bioRxiv. We have combined addressing this with addressing a previous comment, as mentioned above, with a section in the Discussion on Page 16, Line 434.

In Appendix 2, the authors suggest that a major limitation of optogenetics and chemogenetic in- activation methods is the lack of rhesus-compatible constructs. However, several viral constructs have successful implementation in rhesus monkeys so far (e.g. Galvan A, Stauffer WR, Acker L, El-Shamayleh Y, Inoue KI, Ohayon S, Schmid MC. Nonhuman primate optogenetics: recent advances and future directions. Journal of Neuroscience. 2017 Nov 8;37(45):10894-903; Tremblay et al, Neuron 2020)

We thank the reviewer for pointing us to these papers. We have added a more thorough description of what we meant by lack of rhesus-compatible constructs in that Appendix.

“However, other challenges exist with using optogenetics as an inactivation method in nonhu- man primates, including difficulty reliably affecting behavior [12]. While several constructs for rhesus macaques have been developed [13], [14], reports of successfully inducing be- havioral effects have a small effect size and are less numerous than might be expected [12], and several null results have been published [15]–[17]. Other remaining challenges include the need to develop a head-mounted, battery powered light delivery system for multi-day delivery of light and difficulty integrating illumination with simultaneous chronic neuro- electrophysiology.”

For Figure 5b, only pairwise comparison results from monkey U (L11-14) are shown. It is unclear why such results from monkey H were shown in Figure 5a but not in 5b.

We thank the reviewer for pointing out this unconventional one monkey result. As described in the original submission, we previously omitted Monkey H from the analysis in Figure 5b (now Figure 7) since some of the lesions were closely spaced together, preventing well defined pre- and post- lesion rates of turnover. Never-the-less we have included Monkey H in all the revised analysis and believe even the less cleanly separated data shows useful indications of neuron loss or silencing evoked by the lesion.

Behavioral data (during a motor task) from the awake behaving monkeys (U and H) would greatly strengthen the claim that this lesioning method is capable of creating a behavioral effect and can be adopted to study the relationship between neural function and behavior outcomes.

While we are grateful for the reviewer’s interest in the application of our lesioning technique to studies involving behavior, a behavioral analysis of the effects of our electrolytic lesions falls be- yond the scope of this Tools and Resources manuscript. We would also like to point out that we do not claim that we have achieved a behavioral deficit in this manuscript.

Figure 2 would benefit from an illustration of the Utah array placement and the location of the sites used for lesioning. The authors can either overlay the illustrations on the current ex-vivo and histology images or create a separate schematic to demonstrate that for the readers. Also, Figure 2B needs to be replaced with one without the folds to avoid confusion for the readers.

We have added Figure 2 - figure supplement 1, which shows both the location within the Utah array of the two electrodes used to create the lesions as well as the relative size of the surface area of the lesion and the array. Unfortunately, as the lesion was created under the array, the exact location of the array relative to the lesion is unknown.

As mentioned above, Figure 2B is the only histological image from that lesion. We hope that the markings in the image as well as the caption sufficiently explain that the folds are unrelated to the lesion itself.

Figure 3, the conical region is not well delineated. Data across animals and lesion volume with respect to different parameters should be included.

We have included a supplemental figure, Figure 3 - figure supplement 1, where we have used a dashed white line to clearly indicate the area of damaged parenchyma, in case it was not clear in Figure 3a. We have also added volume estimates from lesions across animals and different param- eters. The ex vivo estimates are shown in Figure 4 and the in vivo estimates are shown in Figure 5.

Figure 4: it is not clear what is being communicated, and where the voltage traces are from.

We thank the reviewer for noting this confusion. We have added some lines in the text to explain what the voltage traces show, both in the caption to Fig. 6 and in the text on Page 7, Line 238.

“Traces only capture the values while the lesioning device was turned on (45 seconds for most lesions and 50 seconds for lesion H200120). (A) Voltage traces. Discontinuity at the beginning of the traces indicates transient voltages that were too rapid to be captured by the voltmeter, lasting between 0.13 and 0.33 s. The fluctuating voltages, especially the rapid in- crease in voltage at the beginning of lesioning, emphasize the importance of using a current source to deliver consistent amounts of current into the brain.”

“The voltage across the microelectrode array fluctuated much more than the current did, em- phasizing that we made the correct choice in using a current source to ensure delivery of consistent amounts of current into the brain (Fig. 6).”

Figure 5: why did the authors choose to use matching units as a measure of the lesion? It is surprising that there are still units on the location that the authors claim to be a lesion. To clarify that it would be helpful to show the location of the lesion in Figure 4a. Also, what can we conclude about the lesion induction when we see units on the lesion electrode? The change in unit match shows that there is a change in the network (although the authors need to show control for that so we know those changes don’t happen due to natural dynamics). It is not clear what is the time duration for pre-pre and post-post (i.e. minutes, seconds, hours). Do these comparisons come from the same time frame or are they coming from two fragments of time for both pre and post- conditions?

Aside from post-mortem histology and tissue assays, there is no good way to confirm neuron loss with chronically implanted electrode arrays in nonhuman primates. Waveforms were chosen as they are the one readily isolated physical measure of the system we are injuring. Although functional measures of activity could indicate neuron loss (topic of following papers), there are many conceivable changes in firing rate patterns that could manifest spuriously as loss, making the estimation of loss even more ambiguous and challenging this way.

We believe the new Figure 7 will make the procedure much more clear, while also providing the control requested by the reviewer, illustrating that new statistical categories of altered waveforms emerge during a lesion, beyond those associated with typical changes in waveform composition within multi-unit recordings seen during recording sample turnover fom healthy animals. We further note that by confining this analysis to four day spans at most, we have limited the impact of daily sample turnover described in the literature (Gallego, 2020).

The time duration for pre-session versus pre-session (pre-post and post-post), is some multiple of the approximate 24 hours between each daily recording session. Therefore, since restricting our- selves to four days separation, between 24 and 96 hours. Spikes are sampled from successful trial periods (so on the order of seconds, compiled into minutes across the whole recording session). Although already described in the main text, these points have been reemphasized in the figure legend.

CNO (line 931) needs to be explained.

We thank the reviewer for this point. We have defined CNO and its relevance in Appendix 2.

“Additionally, chronic inactivation over days may be logistically challenging, as the half life of clozapine N-oxide (CNO, a ligand used to activate DREADD receptors) is on the order of hours.”